# Berberine Effects in Pre-Fibrotic Stages of Non-Alcoholic Fatty Liver Disease—Clinical and Pre-Clinical Overview and Systematic Review of the Literature

**DOI:** 10.3390/ijms25084201

**Published:** 2024-04-10

**Authors:** Florentina Ionita-Radu, Cristina Patoni, Andreea Simona Nancoff, Flavius-Stefan Marin, Laura Gaman, Ana Bucurica, Calin Socol, Mariana Jinga, Madalina Dutu, Sandica Bucurica

**Affiliations:** 1Department of Gastroenterology, Carol Davila University of Medicine and Pharmacy, 020021 Bucharest, Romania; florentina.ionita-radu@umfcd.ro (F.I.-R.); cristina.patoni@drd.umfcd.ro (C.P.); flaviusstefanmarin@gmail.com (F.-S.M.); sandica.bucurica@umfcd.ro (S.B.); 2Department of Gastroenterology, Dr. Carol Davila Central Military Emergency University Hospital, 010242 Bucharest, Romania; andreea-simona.nancoff@rez.umfcd.ro; 3Department of Biochemistry, Carol Davila University of Medicine and Pharmacy, 020021 Bucharest, Romania; laura.gaman@umfcd.ro; 4Faculty of General Medicine, Carol Davila University of Medicine and Pharmacy, 020021 Bucharest, Romania; ana.bucurica@stud.umfcd.ro (A.B.); calin.socol@stud.umfcd.ro (C.S.); 5Department of Anesthesiology and Intensive Care, Carol Davila University of Medicine and Pharmacy, 020021 Bucharest, Romania; 6Department of Anesthesiology and Intensive Care, Dr. Carol Davila Central Military Emergency University Hospital, 010242 Bucharest, Romania

**Keywords:** non-alcoholic fatty liver disease (NAFLD), non-alcoholic steatohepatitis (NASH), berberine, fatty liver, liver steatosis, metabolic dysfunction-associated fatty liver disease (MAFLD)

## Abstract

Non-alcoholic fatty liver disease (NAFLD) is the predominant cause of chronic liver conditions, and its progression is marked by evolution to non-alcoholic steatosis, steatohepatitis, cirrhosis related to non-alcoholic steatohepatitis, and the potential occurrence of hepatocellular carcinoma. In our systematic review, we searched two databases, Medline (via Pubmed Central) and Scopus, from inception to 5 February 2024, and included 73 types of research (nine clinical studies and 64 pre-clinical studies) from 2854 published papers. Our extensive research highlights the impact of Berberine on NAFLD pathophysiology mechanisms, such as Adenosine Monophosphate-Activated Protein Kinase (AMPK), gut dysbiosis, peroxisome proliferator-activated receptor (PPAR), Sirtuins, and inflammasome. Studies involving human subjects showed a measurable reduction of liver fat in addition to improved profiles of serum lipids and hepatic enzymes. While current drugs for NAFLD treatment are either scarce or still in development or launch phases, Berberine presents a promising profile. However, improvements in its formulation are necessary to enhance the bioavailability of this natural substance.

## 1. Introduction

Non-alcoholic fatty liver disease (NAFLD) prevalence has increased in the last three decades by more than 12% and stands as the most prevalent chronic liver condition. Non-alcoholic steatohepatitis (NASH), which represents the more aggressive manifestation of NAFLD, is increasingly recognized as a primary contributor to cirrhosis development, complications associated with cirrhosis, liver cancer, and mortality related to liver disease [1,2].

Metabolic dysfunction-associated fatty liver disease (MAFLD) was recently proposed to replace non-alcoholic fatty liver disease (NAFLD) to acknowledge better that NAFLD manifests as a disorder affecting multiple systems. Additionally, experts have suggested a set of simple positive criteria to diagnose and evaluate individuals for this disease. While NAFLD was an exclusion diagnosis, MAFLD is a diagnosis of inclusion with the presence of hepatic steatosis and one of the three features including (1) overweight/obesity, (2) type 2 diabetes mellitus, or (3) lean or normal weight with evidence of metabolic dysregulation [3].

A meta-analysis reported the global prevalence has increased over the past three decades, from 25.26% (95% CI: 21.59–29.33%) in 1990–2006 to 38.2% (95% CI: 33.7–42.89%) in 2016–2019 [4]. Based on the available evidence, the prevalence of MAFLD varies globally and is influenced by geographical region, ethnicity, sex, and obesity status. The highest occurrence is reported in Europe (55.33%, 95% CI 36.20–73.00%), followed by Asia (36.31%, 95% CI 29.89–43.26%), and the lowest is found in North America (35.99%, 95% CI 30.68–41.66%) [5,6]. In parallel, metabolic dysfunction-associated steatohepatitis (MASH), the developing form of MAFLD which is characterized by the presence of hepatic steatosis, inflammation, and hepatocyte ballooning, is rapidly growing as one of the main contributing factors of cirrhosis, the fastest rising cause of hepatocellular carcinoma, and is also the swiftest increasing indicator for liver transplantation in the United States [7]. Due to the increasing prevalence of obesity and diabetes, it is projected that the burden of MAFLD will rise in the next decade. A recently 2023 published study emphasizes the fact that epidemiological data changes when considering the prevalence of MAFLD (which is a more inclusive diagnosis) versus NAFLD and finds at least 5–10% more MAFLD worldwide reported (30–40%) compared with NAFLD reports of 25–30% [8].

The actual available therapies for NAFLD are still under development and need to be tested. As a result, there is a pressing need to develop new therapeutic strategies to address this unmet need. On the other hand, berberine is a natural compound used for many years as an accessible, versatile herbal therapy, especially in oriental medicine. It has proven hypolipemic and anti-steatotic effects in NAFLD [9,10].

Berberine represents one of the plant’s secondary compounds: Benzyl-iso-quinoline alkaloids (BIAs). These natural alkaloid compounds are found in the outermost layer of stems and roots of multiple therapeutic plant species from the *Berberidaceae* family, genus *Berberis* [10].

The interest in Berberine has increased in recent years due to its multiple benefits, as botanical alkaloids such as Isoquinoline alkaloids have been proven to have versatile medical effects. As poly-pharmacological compounds, alkaloids showed antiphlogistic, antiseptic, antioxidant, pain-relieving, carcinogenetic-inhibiting, antispasmodic, and anti-tussive effects [10,11,12] (Figure 1). Berberine and its metabolites and derivatives have various medical actions. They have a multiorgan distribution, with the utmost concentration at the hepatic level. However, they also have a high action in the renal, muscles, respiratory, cerebral, heart, and pancreas and the least amount in fatty tissue [10].

### 1.1. Diagnosis of Liver Steatosis

The diagnosis of liver steatosis should be sustained in the presence and quantification of the hepatic fat. The diagnosis is based on hepatic biopsy, which is considered an invasive procedure; alternative methods are image-based, non-invasive, or blood-based tests (Figure 2).

The diagnosis of NAFLD should be sustained when there were excluded substantial alcohol intake and additional recognized factors of liver disorder [15,16,17].

#### 1.1.1. Pathology in Pre-Fibrotic Stages of NAFLD

Diagnosing liver steatosis accurately can sometimes be challenging because the gold-standard liver biopsy is an invasive diagnostic procedure susceptible to constraints and adverse outcomes, although pre-fibrotic changes in the liver could be expressed before any histological modifications occur, various tests are proposed for diagnosing and staging NAFLD [18,19]. Still, liver biopsy remains crucial for identifying NASH and remains the sole method capable of accurately distinguishing between non-alcoholic steatosis (NAFL) and non-alcoholic steatohepatitis (NASH). However, it has constraints owing to sampling inconsistencies [20].

The initial liver changes at the tissue-level examination in NAFL consist of more than 5% of hepatocytes with steatosis but without associating one of the intracellular ballooning or fibrosis. The emerging steatohepatitis is characterized by patchy inflammation of the hepatic lobules and variable minimal fibrotic changes [16,21].

The consortium of European pathologists proposed and validated the fatty liver inhibition of progression (FLIP) algorithm, which categorizes cases as NASH, NAFL (non-NASH NAFLD), or non-NAFLD, taking into consideration the presence of steatosis, hepatocyte clarification, or hepatocytes ballooning, and inflammatory changes in liver lobules.

According to the FLIP score, the intracellular content of medium or large-sized lipid vacuoles and the percentage of affected hepatocytes were assessed and classified as grade 0 if there is less than 5% of the cells involved, grade 1 between 5 and 33%, grade 2 between 34 and 66% and grade 3 if there are more than 67% of hepatocytes affected [22] (Figure 2). It is considered non-NAFLD if less than 5% of hepatocytes are involved. Still, if they are present, a minimum grade 1 of one of the other characteristics, such as inflammation of the lobules, ballooning, or steatosis, is considered NASH.

Consecutively, Nascimbeni et al. proposed the Steatosis-Activity-Fibrosis (SAF) score, which evaluated hepatocyte ballooning and lobular inflammation. According to this grading, the histologically severe disease was characterized by a SAF activity score exceeding three and/or the presence of bridging fibrosis or cirrhosis [22].

According to the American Association for the Study of Liver Disease (AASLD), the NAFLD Activity Score (NAS) represents the disease’s activity without considering fibrosis. At the same time, Steatosis-Activity-Fibrosis quantifies steatosis, activity (NAS 0 to 8 points), and fibrosis [14]. The rationale for a scoring system is justified due to significant variability in biopsy specimen reports and feasibility [20].

When considering a liver biopsy for NAFLD, it is essential to account for the cost-efficiency, the quality of the specimen, the pathologist’s consistency in the interpretation or the level of expertise, the heterogenous damage of the liver parenchyma, the invasiveness of the procedure, and, nevertheless, the willingness of the patient to undergo the biopsy [23].

#### 1.1.2. Imagistic Studies in Pre-Fibrotic Stages in NAFLD

The non-invasive methods aroused interest for the non-invasive diagnosis of NAFLD, and these comprise ultrasound technique, vibration-controlled transient elastography FibroScan, and magnetic resonance spectroscopy or magnetic resonance imaging-derived proton density fat fraction and serum biomarkers [3,24] (Figure 2).

Regarding the reliability of ultrasound B-mode for assessing hepatic steatosis, it may be prone to subjectivity and operator dependency. It relies on estimating liver brightness, discerning echogenicity differences between the liver and kidney, and visualizing liver vessels. Its sensitivity is lower than 70% for moderate steatosis but higher for severe steatosis [25,26]. Although variations of interpretation are admitted between examinations of the same operator or between different operators, B-mode ultrasound remains a safe procedure, non-irradiating, affordable, reproducible, and a first-line assessment method when the fatty liver is suspected [16,26].

The Hamaguchi ultrasound score was proposed to improve the accuracy of hepatic steatosis assessment. It considers the depth attenuation, the vessel fading image, the liver brightness, and the contrast between liver and kidney parenchyma. Although not widely used, it may be a good surrogate parameter when Magnetic Resonance Spectroscopy (MRS) is unavailable [26,27,28].

Another ultrasound-based technique proposed by the World Federation of Ultrasound in Medicine and Biology (WFUMB) is the Acoustic Radiation Force Impulse (ARFI) method, which uses the shear wave technique by impelling a longer beam of ultrasound pulses. Also, the ARFI methods may be categorized as point Shear Wave Elastography (p-SWE) or 2-D Shear Wave Elastography (2-D SWE), and the measured stiffness can be converted into kPa, the same as the CAP FibroScan [29]. Furthermore, a method that quantifies ultrasound beam attenuation in liver tissue and enables the automated extraction of image artifacts from the assessment of liver steatosis was developed, and this method shows a strong correlation with the more accurate MRS assessment [26].

Controlled attenuation parameter (CAP) using transient elastography method FibroScan (Echosens, Paris, France) showed good performance compared with biopsies results, especially in identifying steatosis grade 1 and 3 and related to ultrasound assessment CAP was more feasible in steatosis cases [30,31]. The Youden used cut-offs varied between 302 dB/m for grade 1 and 337 dB/m for grade 3 of steatosis [30]. The CAP score grades steatosis as grade 0 for less than 278 dB/m, between 278 and 301 dB/m for grade 1, 303 and 337 for grade 2, and more than 337 for grade 3 (Figure 2). The CAP assesses quantitative modification of the ultrasound beam as it travels through the hepatic tissue and has lower inter-operator variability. Factors that could influence the measurements were the female sex, diabetes mellitus, obesity, or metabolic syndrome [26,32].

The Magnetic Resonance-based technique for liver fat estimation is the Magnetic Resonance Imaging Proton-Density Fat Fraction (MRI-PDFF), which provides accurate estimation and is widely used in clinical trials. MRI assessment has proven to be even more sensitive than histological findings in detecting changes in fatty liver after 24 weeks of interventional studies. Still, it does not provide information regarding fibrosis’s inflammatory status or alleviation [33].

Another MRI method is Chemical-Shift Encoded MRI (CSE-MRI), and both procedures detect the movement of free water and triglycerides, unlike the standard MRI, which cannot perceive the structural lipids [34]. CSE-MRI measures the proportional quantity of water and fat signals originating from the liver, while MRI-PDFF assesses the proportion of mobile protons density attributed to triglycerides, relative to the combined density of protons from both mobile triglycerides and mobile water, as a percentage (%) ranging from 0 to 100%, represents an essential feature of the tissue, indicating the concentration of mobile triglycerides within it [34].

Regarding Positron Emission Tomography–Computed tomography scan (PET/CT) usefulness in staging and diagnosing NAFLD, there are needed standards because the relationship between body mass index and correction of standardized uptake value (SUV) of [18F]2-fluoro-2-deoxy-D-glucose (18F-FDG) dosage is not established in non-malignant cases [35,36]. Alternative PET/CT tracers proposed for liver steatosis diagnosing are 14 (R, S)-[18F] fluoro-6-thia-Heptadecanoic acid (18F-FTHA), 11C-Palmitate, and 11C-Acetate, but still need validation and standardization [37].

#### 1.1.3. Serum Markers in NAFLD

For the time being, liver biopsy is considered to be the gold standard when it comes to NAFLD accurate diagnosis. However, carrying out biopsies every time steatotic liver disease is suspected remains controversial, and that is why tedious research has been directed towards less invasive tests that can offer a better alternative [38]. These new research areas consist of different markers and scoring systems that could potentially replace invasive or expensive methods in the near future. While available tests for the fibrotic stages may be helpful in clinical practice, the serum panels or biomarkers need further validation for NASH. Regarding apoptosis markers cytokeratin (CK)-18, a hepatocyte apoptosis fragment was the most studied for NAFLD diagnosis [39]. Even though CK-18 is unavailable for usual practice and more feasible for trials [18,29], recent studies showed a promising result regarding NAFLD diagnosis, especially in combination with FIB-4 (fibrosis 4 index) test and MACK-3 test (a new blood test consisting of Homeostatic Model Assessment-Index Insulin Resistance (HOMA-IR), aspartate aminotransferase (AST), and CK18 levels) [40].

Furthermore, the perspective of gene-based markers seems attractive. Similarly to CK18, the clinical utility is feeble for the NASH score that relies on the Patatin-like phospholipase domain-containing protein 3 (PNPLA3) genotype or tests for microRNA profiling [29]. Zeng Y. et al. (2022) proved in their study that the so-called omics biomarkers PNPLA3, TM6SF2 (transmembrane 6 superfamily 2 human gene), different types of microRNAs such as miRNA-122, miRNA-199, miRNA-34a-5p or extracellular vesicles could open new doors to future diagnostic methods for steatotic liver disease. Still, these need further study [39].

In addition to all of these, there are several tests proposed to diagnose NAFLD, the six most studied being the fatty liver index (FLI), AST, platelet ratio index (APRI), FIB-4 index, AST/ALT 4 ratio, Bard score, and NAFLD fibrosis score (NFS) [41]. Most of these co-scoring tests consist of a combination of serum biomarkers and anthropomorphic parameters such as age, AST, gamma-glutamyl transferase (GGT) levels, platelet count, serum albumin levels, international normalized ratio (INR), impaired fasting glycemia or presence of diabetes, and body mass index [41,42]. Although they were demonstrated to be effective in excluding severe cases of fibrosis, none of them have proven to differentiate between the lower and more advanced stages of liver fibrosis [43]. Another promising panel blood-based biomarker appears to be NIS-4, which includes markers representative of NASH, such as α2-macroglobulin, miR-34a-5p, YKL-40, and hemoglobin A1c (HbA1c) [14,41]. This suggests that it could be a valuable tool in identifying features of NASH [14,41]. No established test, marker, or algorithm can accurately diagnose NAFLD; recent research has found new and innovative ways through which we hopefully will replace liver biopsy as the sole precise method for steatosis liver disease diagnosis.

### 1.2. Pathophysiology and Molecular Signaling in NAFLD

The progression from pre-fibrotic to fibrotic stages in NAFLD is estimated to occur in more than 40% of cases, with an annual progression of around 1% [7]. In pre-clinical studies, the changes observed after a high-fat and carbohydrate-based diet showed progressive alterations at the molecular level in the liver, gradually increasing from lobular to portal inflammation for 20 weeks (from week 8 to week 27). There was an increased expression of mRNA and protein of chemokine monocyte chemoattractant protein 1 (MCP-1), which is responsible for fatty transformation and is upregulated during fibrogenesis [44]. In the same study by Ganz et al., a specific dynamic of inflammation markers was observed, with Tumor Necrosis Factor α (TNFα) mRNA and Interleukin-1β (IL-1β) mRNA initially overexpressed at the liver tissue level and subsequently in the serum, through nuclear factor kB (NF-kB) progressive upregulated pathway [44].

The inflammation cascade is triggered by the excessive intake of fats and carbohydrates through activation of the P2X7 receptor NLRP3 and TLR. Nevertheless, the well-known gut-lipopolysaccharides trigger the immune Kupffer cells and initiate the other inflammation cycle [45]. Through this course, there are activated M2 macrophages and Kupffer cell markers such as Cluster of Differentiation (CD) 11b, CD 68, and CD 163, which are overexpressed in the pre-fibrotic stage of NAFLD, and these tend to fade when fibrogenesis is initiated and switch to M1 macrophages activation [44].

There are progressive steps in fatty liver transformation, and these steps involve the process of liver active uptake of fatty acids, which is dependent on transporters such as Fatty transport protein (FATP)-2, FATP-5, CD36, and membranous caveolins [9]. The downregulation of FATP-2 and FATP-5 in pre-clinical studies was related to alleviating fatty burden into the liver tissue, lowering fatty acids transport into hepatocytes, and decreasing triglycerides at this level. At the same time, CD36 favored the access of long-chain fatty acids through liver X and pregnane X hepatic receptors and peroxisome proliferator-activated receptor (PPAR)-γ [9]. At the cellular level, the fatty acids are directed to intracellular organelles through liver fatty acid binding proteins (FABP) 1, which may vary during the progress of steatosis [9]. Further steps include endogenous lipid synthesis, the oxidative process of fatty acids, and the release of very low-density lipoproteins (VLDL). Endogenous lipid synthesis is based on converting acetyl coenzyme A into palmitate, which, after biochemical complex processing, is transformed into triglycerides or VLDL. Nevertheless, palmitate showed inflammatory effects and promotes steatohepatitis [9].

In a recent study by Moore et al., it was shown that oxidative stress and altered lipid oxidation are linked to hepatocyte mitochondrial morphological and functional impairment in steatohepatitis, further compounded by a decreased presence of oxidative phosphorylation (OXPHOS) proteins as it progresses to fibrosis [46]. Essentially, the accumulation or overproduction of lipids surpasses their exportation or oxidation, leading to the release of surplus lipids as VLDLs, with limitation when the liver’s fat content reaches 10% [47].

Fibroblast growth factor 19 (FGF-19) is a protein with a hormonal function that regulates bile acid secretion after daytime meals. The enterocytes of the ileum release FGF-19, which is regulated by the farnesoid X receptor (FXR) in response to changes in bile acid flux. Subsequently, the FGFR4-KLB complex on hepatocyte membranes is activated [48,49]. FGF-19 plays a double role in maintaining the balance of lipid metabolism [50]. It reduces the lipids by boosting the metabolism through FGFR1 c and FGRFR4 and reduces the bile acids production in the liver via Cyp7A1 and Cyp8b1 enzymes transcriptional rates [48,51]. On the contrary, the FGF21 protein exerts multi-metabolic hormonal action. It exhibits high expression at hepatic levels, where it is secreted in reaction to elevated glucose, increased free fatty acids, and diminished amino acid availability. It governs energy, glucose, and lipid balance through its effects on the central nervous system and adipose tissue [52].

The AMP-activated protein kinase pathway (AMP) is a major intracellular complex that upregulates metabolism, energy homeostasis, and cellular growth. Furthermore, the AMPK signaling pathway exhibits heterogeneous effects through various molecular reactions, which recently became of interest regarding NAFLD [53].

Sirtuin 3 (SIRT3) is a mitochondrial protein mainly involved in the oxidative stress response. One of the major causes of metabolic imbalances associated with NAFLD is the increase of reactive oxygen species synthesis. SIRT is abundant in tissues with intense metabolic activity. One of the most important roles of this protein is modulating fatty acid oxidation induced by oxidative stress [54].

Peroxisome proliferator-activated receptor-Υ (PPAR-Υ) is a nuclear receptor that plays a key role in lipid metabolism and inflammation response. It is well known that the expression of PPAR-Υ is heightened in adipose tissue, which is involved in adipogenesis, fatty acid synthesis, and glucose homeostasis, and inhibition of liver PPAR-Υ prevents the evolution towards fatty liver [55,56].

### 1.3. Actual Therapeutic Agents in NAFLD

The perspective of multisystemic metabolic disorder, which encompasses NAFLD as MAFLD when it coexists with other metabolic alterations or diseases such as type II diabetes mellitus or obesity, suggests that first-line intervention should be diet and a healthy lifestyle. Conversely, there is a likelihood that a subset of patients previously diagnosed with NAFLD may not meet the criteria for MAFLD, exhibit apparent alcohol consumption, or have known unusual causes of liver disease and might fail to identify a sub-population with steatohepatitis not related to metabolic factors and with considerable fibrosis [57]. This may represent a different perspective and should stimulate novel findings regarding the etiopathogenesis, categorization, and management of fatty liver disease [3]. Lifestyle changes are challenging at the individual level and require long-term upholding, necessitating the development of medical treatments [58].

The main reason for the delayed development of medical treatment for NAFLD is the difficulty in pairing an accurate pre-clinical experimental prototype to resemble the human liver, most used for experimental studies being mice, rats, cell cultures, and guinea pigs, which are the most feasible [9].

The most recently approved molecule for NASH treatment is Resmetirom, a thyroid hormone receptor beta (THR-β) selective agonist, with per os administration [59,60]. Resmetirom proved to decrease the hepatic fat load observable on MRI-PDFF, which was corroborated with improved biochemical markers of hepatic profile and fibrosis and a lower value in serum lipid profile [61]. The MAESTRO-clinical trial proceeded to MAESTRO-NAFLD-1 and MAESTRO-NAFLD-OLE phase 3 clinical trials to assess further and establish the safety drug profile of Resmetirom in order to obtain Food and Drug Administration (FDA) approval as the first drug for the treatment of NASH with or without fibrosis [61,62]. Other research focused on PPAR-Υ agonist pioglitazone, which induced weight gain as an adverse event [9], and farnesoid X receptor agonist obeticholic acid, which was associated with serum LDL cholesterol higher values and exacerbated cutaneous pruritus [9]. Another PPAR agonist used in studies that included patients with NAFLD is saroglitazar, alongside the improvement of alanine aminotransferase (ALT) and dyslipidemia, showed a decreased liver fat content of up to 45% after 16 weeks [58].

Potential treatments for NAFLD are notable, including FGF-19 and FGF-21 analogs. Aldafermin, a non-mitogenic analog of FGF-19, demonstrated improvements in MRI-PDFF values of liver fat or histologic NAS score associated with lower serum levels of lipids and liver enzymes after 12 weeks of treatment in patients with NASH. Studies also observed decreasing trends in serum fibrotic markers, such as pro-peptide type III collagen (Pro-C3) [49].

The use of Pegbelfermin and Efruxifermin, both FGF-21 analogs, resulted in a significant reduction in hepatic fat content in patients diagnosed with histological NASH [63,64]. In the CENTAUR and AURORA studies, the dual chemokine receptor antagonist 2/5 (CCR2/CCR 5 antagonist)-cenicriviroc improved the fibrosis stages without exerting effects on NASH [65,66]. In a phase 2b trial conducted by Loomba et al., the administration of a selective inhibitor of apoptosis signal-regulating kinase 1 (ASK-1), selonsertib, resulted in more than one stage of fibrosis regression in patients with NASH and F2-F3 fibrosis scores [67].

A significant aspect of NAFLD, which could potentially benefit from future molecular interventions, is the presence of reduced markers of mitochondrial turnover preceding decreased hepatic fatty acid oxidation and increased reactive oxygen species (ROS) production. This suggests that these reductions may initiate further deterioration of mitochondria and subsequent progression to NASH. These data collectively highlight a critical and novel role for fatty acid oxidation and mitochondrial turnover in the liver and indicate potential future targets for preventing and treating NAFLD/NASH in humans [46].

## 2. Methods

The aim of our review is to highlight berberine’s effects as a natural and available compound on different metabolic processes, with emphasis on its different molecular pathways and their outcomes regarding NAFLD.

We systematically searched two databases, Medline (via Pubmed Central) and Scopus, from inception to 5 February 2024. Our search contained two concepts, one related to berberine and the other related to fatty liver. The following search key was used: ((berberine [MeSH Terms]) OR berberi*) AND (liver[MeSH Terms] OR “fatty liver” [MeSH Terms] OR liver OR NAFLD OR NASH OR MAFLD). In addition to using automated search methods, we also manually searched for citations from reviews related to similar topics. The review only considered published articles. No specific protocol was created for this review.

The articles yielded by the advanced search were downloaded into Endnote X (Clarivate Analytics, Philadelphia, PA, USA). Duplicates were automatically removed and then manually removed afterward. Two independent researchers (SB and CP) conducted the selection, and a third party (AN) solved disagreements (Figure 3).

We considered eligible nonhuman studies on rats, mice, or cell lines and clinical studies.

We excluded reviews, case reports, studies that did not state how they established metabolic-associated fatty liver disease diagnosis, drug-induced, ischemic, viral, or autoimmune causes of liver injury, and alcoholic fatty liver disease.

Two independent evaluators (AN and AB) extracted data. The data included details such as the first author, publication year, total number of trial participants, intervention measures used, berberine dose, and intervention duration.

The primary outcome comprised the NAFLD diagnosis and effects of berberine on NAFLD, and the secondary outcome was represented by effects on biochemical markers, lipid profile, and liver enzymatic profile related to NAFLD, about Berberine (BBR) and its impact on NAFLD, along with liver serum enzymes and serum lipids, as well as metabolic disorders associated with NAFLD such as type II diabetes mellitus or obesity; we searched for relevant pre-clinical and clinical studies.

## 3. Results

In the context of interventional studies in NAFLD, there is a focus on potential therapies with anti-steatosis effects. This is particularly evident when there is substantial evidence of a reduction in hepatic fat content during phases I or II, alongside secondary endpoints such as serum liver enzymes or lipid profile [34]. The established association of an interventional measure with weight loss and the reduction of liver fat content without observed toxicity facilitated the progression to phase 3 trials [34].

### 3.1. Clinical Studies on BBR and NAFLD

The studies involving human subjects relied on objective quantification of the amount of fat in the liver using MRS, MRI-PDFF, ultrasonography, and ^18^F-FDG PET/CT before and after administration of a form of BBR formula and also measured the lipidic and hepatic biochemical profiles (Table 1).

The Berberine action in humans is related to its biological availability, which greatly influences the results. According to the Biopharmaceutics Classification System (BCS), berberine could be classified as BCS Class II and IV. This classification is attributed to berberine’s biological effects being influenced by its bioavailability, which remains lower than 0.01 within 48 h. This limitation is primarily due to its restricted solubility and slow dissolution in water [77]. Also, berberine has decreased absorption in the intestines, showing that less than one-third is absorbed at the intestinal level in animal studies [78].

This is an impediment because the easiest and optimal administration is the oral route, the parenteral administration being related to adverse toxicity of berberine, uneven diffusion, low cellular internalization, and fast drug elimination [78,79]. Another factor that affects the berberine’s biological availability is the intervention of efflux membrane transporter P-glycoprotein, which limits the cellular uptake of the xenobiotics in the gastrointestinal tract [77].

The methods developed to increase berberine’s availability are delivery by nanosized carriers with both inorganic and organic components such as polymers, lipids, gold, or magnetic porous nanomaterials [78]. In the study of Kohli K. et al., the non-toxic, biologic-compatible polymers used were chitosan and alginate, and the results showed a remarkably increased bioavailability of berberine [77]. The nanotechnology medication formulation with encapsulating micelles as transporters enables the solubilization of hydrophobic BBR molecules without altering the pharmacological composition [80]. BBR as a hydrophobic core enveloped in a thin shell of hydrophilic micelle segments (BBR-incorporated micelle solution) proved to be feasible with an 80% rate of success and an increased effect of BBR compared with standard formulation administration in pre-clinical studies [81]. Another proposed formula used poly-lactic acid (PLA) as a bioactive material configuration stabilized by the coaxial electro-spraying procedure for obtaining BBR nano-layering [82]. In this in vivo and in vitro study, the results showed a better internalization of BBR-loaded poly-lactic acid nanoparticles at the cellular level compared with standard BBR, with a better release and action on neoplastic transformed cells [82]. Other proposed delivery systems to improve the bioavailability and to decrease the degradation of BBR were carriers such as lipid vesicles, non-ionic surfactant vesicles, dendritic nanoparticles, nanosilver or nanogold particles, ultrafine emulsions or ethosomes (ethanol-based particles) [83].

### 3.2. Pre-Clinical Studies of Berberine Effects in Experimental Models

To better understand the various effects of BBR, we tried to summarize the multiple pathways in which this compound exerts its properties. BBR is a versatile alkaloid that is involved in a wide range of molecular and chemical reactions associated with fatty liver disease pathophysiology. In the following paragraphs, we presented these molecular targets, how BBR acts on them, and their metabolic effects (Figure 4, Table 2, Table 3 and Table 4, Appendix A).

Recent studies conducted on both animal model experiments and different types of cell cultures showed that BBR modulated lipid metabolism and has anti-steatotic and anti-inflammatory effects. The studies on various experimental models described BBR effects on multiple pathways involved in the progression to NAFLD, and the major ones comprise activation of the SIRT3 pathway, upregulation of SIRT1, activation of AMPK pathway, suppression of NLRP3 pathway, and act as PPAR-Υ agonist (Figure 4, Appendix A).

As we found, BBR plays a complex role when it comes to modulating different metabolic processes, and through the studies that we analyzed, it was shown that BBR also has different pharmacological effects [147,148]. The existent literature consists of different experimental models ranging from different species of rats and mice to various cell cultures. When we talk about animal experiments, BBR has proved to reduce anthropomorphic features of rodents, improve control of lipid and glucose metabolism, and target inflammation by modulating different genes and pathways involved in stress oxidative response. Moreover, in cell cultures, BBR proved to have an effective anti-steatotic property, showing a decrease in the accumulation of lipids or an improvement in NAFLD-induced changes in different cell lines. In the last few years, researchers have been keen to find the relationship between different inflammatory pathways and their contribution to NAFLD’s progression [138]. Some studies we analyzed found that mitochondria may play a more significant role in stress oxidative response than was known. Xu X. et al. (2019), Sun Y. et al. (2017), Rafiei H. et al. (2017), and Teodoro JS. et al. (2013) all proved through their research that BBR modulated essential mitochondrial respiratory complex subunits, especially I and III mitochondrial respiratory chains, and by doing so, managed to decrease the levels of ROS and lower the inflammation both in vivo and in vitro [98,110,130,144]. To better understand NAFLD’s pathophysiology, it is necessary to assess how changes in different species of gut microbiota promote fat liver accumulation. [45,127]. Genomic DNA analysis of gut microbiota proved to be a field of interest for Chen D. et al. (2023) and Zhou LM. et al. (2023), Dai Y. et al. (2022), Yang S. et al. (2022), Li H. et al. (2022), Cossiga V. et al. (2021), and Shu X. et al. (2021), who managed to reveal that BBR not only blocks bacterial translocation but also rebalances gut microbiota by stimulating beneficial bacteria in animal model experiments such as *Atopobiaceae*, *Brevibacterium, Christensenellaceae*, *Coriobacteriales*, *Papillibacter*, *Pygmaiobacter*, *Rikenellaceae RC9* and *Prevotella* in animal model experiments [84,85,87,90,118,121,122].

Another relatively new pathway that seems to be involved in NAFLD’s pathophysiology that has been studied recently is the one related to Clock and Bmal1 genes that are involved in circadian rhythm control [115]. Ye C. et al. (2023) found an interesting association between these genes and BBR treatment. These genes are important for metabolic and redox homeostasis; thus, the disruption of their activity acts as a trigger for inflammatory response. After BBR treatment, by downregulating the activity of Bmal1 and Clock genes, the inflammation was lowered, and thus, the metabolic balance of different molecular reactions was restored [111,115]. With aid from all of these new findings in the field of NAFLD pathology, there is hope for finding a suitable treatment to prevent the many complications associated with this disease.

## 4. Discussion

We want to bring forward new data that we collected from the studies that we analyzed. In our systematic review, we included novel results regarding Berberine’s usage in NAFDL and its beneficial effects on this category of patients; thus, we encompass recent studies regarding BBR treatment in steatotic liver disease. Even though most of the evidence-based on both human and animal experiments needs further testing, BBR opens a new perspective for NAFLD’s treatment.

### 4.1. Effects of Berberine in NAFLD Clinical Studies

In their phase 2 study, Harrison et al. showed that BUDCA treatment might be an option for treating patients with NAFLD and other characteristics of metabolic syndrome. Not only did they find a significant reduction in liver fat content after high-dose BUDCA treatment, but this combination of berberine and ursodeoxycholate also significantly impacted glucose profile. The glycemic profile was better controlled by a higher dose of BUDCA treatment, as seen by the reduction in HbA1c mean levels in this group (two times more than in low-dose BUDCA treatment, 0.6% vs. 0.3%). No changes in fasting blood glucose, HOMA-IR, and insulin levels were detected. BUDCA high-dose treatment was shown to have a significant impact on weight loss. Subjects lost more than twice the weight lost by those on lower dose BUDCA treatment (−3.5% vs. −1.6%, *p* = 0.012). Even though BUDCA had a significant impact on liver function, surprisingly, there was no change in HDL-c, and TG levels were better controlled by lower doses of BUDCA (a decrease of 41 mg/dL in this group vs. 24 mg/dL in 1000 mg BUDCA treatment). Also, the authors stated that the formula they used in this study, HTD1801 (berberine ursodeoxycholate or BUDCA), distinguishes itself from these agents in both structure and function and is different from other investigational agents used in NAFLD treatment, such as obeticholic acid, functioning as a farnesoid X receptor (FXR) agonist bile acid, aramchol, a liver-targeted SCD-1 inhibitor, and the thyroid hormone β agonist, resmetirom [68,149]. Another factor in the NAFLD pathogenesis pathway is FGF-19, which acts as a hormonal regulator. However, as shown by Harrison et al., BUDCA treatment had no significant impact on serum FGF-19 or fibrotic indicators such as propeptide type III collagen (Pro-C3) and Enhanced Liver Fibrosis (ELF) score [68].

In the study of Yan et al., although the BBR group had similar changes in hepatic fat content with the pioglitazone group (the positive control reference for this study), the berberine cohort exhibited notably greater reductions in liver fat content compared to those in the LSI group, alongside more pronounced decreases in blood glucose levels, triglycerides, and cholesterol. Moreover, similar results concerning glucose homeostasis and liver enzyme levels were discovered between the BBR-treated group and the Pioglitazone group, but BBR was more efficient in reducing body weight and improving serum lipid profile [69].

Eleven subjects were randomly selected, and their blood and urine were further tested to evaluate the safety and bioavailability of BBR from the LIS alone and BBR-treated groups. The results showed that BBR presented good absorption, and regarding the serum BBR and its metabolites, the concentration was 50 times lower compared with the hepatic uptake. This suggests that BBR may activate different hepatic genes involved in lipidic and glucose metabolism [69].

In the study of Herrera et al., the results were relevant, especially for the changes in the body mass index, which were greater in the BBR-treated group, and lower serum levels of triglycerides, total cholesterol, and LDL-c, alongside a better glycemic profile in these patients. There were no significant changes in ALT or AST after 20 weeks of BBR treatment [70]. Interestingly, the study reported the impact of BBR in tumor necrosis factor-α (TNF-α) and highly sensitive C-reactive protein (hs-CRP) [70].

In the study by Nejati et al., it was reported that not only serum triglyceride (TG) levels were not improved after treatment with BBR, but also TG levels were reduced to a greater extent in the control group compared to the BBR-treated group. Furthermore, no significant decrease was observed in either group’s total cholesterol (TC) or TG levels (*p* = 0.326 and *p* = 0.464, respectively). In addition, BBR proved not to significantly impact liver enzyme profile, fasting blood glucose, body weight, and BMI among subjects. The differences between the other studies with BBR as an interventional study agent is the formulation and dosage of BBR in this study, as the patients received 6.25 g daily prepared by boiling 100 g of dried BBR in 5000 mL of water until reduced to 4000 mL through boiling [73]. Since the bioavailability of BBR is dependent on the formulation, this preparation method may influence the concentration and absorption of BBR and explain the study’s results.

The research conducted by Chang et al. demonstrated that BBR lowered liver fat content, as quantified by 1HMRS, by an additional 7% compared to cases that only underwent lifestyle changes. Both groups had no statistically relevant differences regarding liver enzyme levels [71]. BBR caused a more significant reduction in HFC than LSI alone. Furthermore, BBR treatment was associated with significant improvement concerning many areas: ameliorated anthropomorphic features (lower body weight, smaller waist circumference, BMI—*p* < 0.01), decrease of liver fat burden—*p* < 0.01; improved glucose homeostasis (lower blood glucose, better control on glucose metabolism—HbA1c—*p* < 0.01), and better control on lipidic metabolism (lower serum levels of cholesterol, triglycerides, LDL-c, apoA/B, LP(a)—*p* < 0.05). In their study, Chang et al. observed that BBR, through its ability to modulate different types of lipids, had similar effects, with just lifestyle changes, on lipid metabolism, except the ceramides, which were influenced solely by BBR treatment [71]. As ceramides are involved in the NAFLD pathogenic pathways, their inhibition may prevent the evolution of NASH [71].

Insulin sensitivity was improved by decreasing the visceral and subcutaneous adipose tissue after BBR treatment. In addition to this, HOMA-IR was also improved in subjects that underwent BBR treatment. To further test these findings among humans, the authors proposed an animal model using diet-induced obese mice. BBR treatment was also associated with changes in brown adipose tissue and thermogenesis and increased energy expenditure, as shown on ^18^F-FDG PET/CT. Furthermore, there was also an improvement in glucose metabolism in the animal model group [76].

In their study, Cossiga V. et al. (2019) tried to evaluate the effects of plant extracts consisting of Berberis aristata, Elaeis guineensis, and Coffea canephora on NAFLD patients [75]. At the end of their experiment, they assessed the liver fat content by measuring mean CAP, and they found that treatment with these plant extracts significantly decreased mean CAP from 291.6 ± 39.2 dB/m to 251.3 ± 41.5 dB/m with a *p*-value < 0.01. This further proves that BBR could be an effective treatment for fatty liver disease. Moreover, there was a significant improvement in glucose metabolism in the treated group compared to the placebo group, as shown by lower levels of blood glucose levels (*p* < 0.001), an improvement in HOMA-IR index (*p* < 0.001), as well as lower levels of insulin (*p* < 0.01). They found no significant changes in the liver enzyme and serum lipid profiles. Overall, their study by Cossiga V. et al. showed that BBR can potentially exert a hepatoprotective effect and improve glucose homeostasis associated with NAFLD [75].

Another study conducted on patients with NAFLD treated with BBR by Yan H. et al. (2021) [74] studied the effects of pioglitazone, BBR, and lifestyle changes in men and women. Their results showed that pioglitazone intervention was superior for improving both lipid and glucose metabolism, mainly in women; regarding liver fat content in BBR-treated groups, women showed a decreased liver fat content when compared to the lifestyle-only group (*p* = 0.020), while men have a better response in BBR group compared to pioglitazone treated group (*p* = 0.007). Moreover, BBR was shown to have no significant interaction regarding efficacity and gender among the studied groups [74].

Regarding side effects, BBR proved to be relatively safe. In all human studies, patients were regularly evaluated for safety reasons, and diarrhea was one of the frequent adverse reactions reported by the subjects. Others complained of nausea, anorexia, dyspepsia, abdominal pain or distention, constipation, and mainly gastrointestinal symptoms. Overall, no life-threatening reaction was observed throughout the administration of Berberine compounds [68,70].

### 4.2. Effects of Berberine and Its Mechanisms—Experimental Models

The utilization of experimental models may be restricted by accurately representing human dietary characteristics, such as replicating the Western diet or detrimental high-fat, high-carbohydrate diets in animal models [9]. Another aspect is that BBR undergoes rapid metabolism in rats and swiftly reaches hepatic tissue [78,81]. In the following paragraphs, we try to summarize BBR’s ways of action and its beneficial properties.

#### 4.2.1. AMPK-Kinase Pathways

Zhao J. et al. proved in their study that BBR combined with metformin through AMPK pathway activation managed to decrease the production of pro-inflammatory cytokines. Moreover, this co-administration of these agents proved superior for ameliorating steatosis and dyslipidemia [101]. Li QP et al. and Qiang et al. also studied the activation of the AMP pathway. There, they found that oxy-berberine (OBB), a metabolite of BBR, and Demethyl-berberine (DMB), an analog of BBR, both work in a similar way as BBR through the AMP-protein kinase route. The effects included improved liver and serum lipid profile, histological improvement, decreased oxidative stress response, and protection against liver injury. Moreover, they demonstrated that OBB was superior to BBR in terms of its hepatoprotective and lipid-lowering properties [89,133].

Furthermore, BBR, by activating the AMPK pathway, proved to downregulate the expression of SREBP-1c both in vivo and in vitro [97,102,127,141]. SREBP-1c (transcription factor sterol regulatory element binding protein-1c) is a key protein that promotes a series of genes responsible for fatty acid production and other genes involved in glucose homeostasis [150]. Overall, BBR boosts AMPK activity and further proves to be a potent anti-steatotic agent that enhances systemic insulin sensibility.

#### 4.2.2. Gut Microbiota

Dysbiosis is known to be another key factor involved in NAFLD’s pathophysiology and progression. By disrupting the intestinal barrier, a large number of bacteria from the intestines, bacterial components, and various other stimuli enter the portal flow, reach the liver, and eventually set off an inflammatory response [45,127,151]. Moreover, this constant exposure to inflammation and large quantities of pro-inflammatory cytokines will participate in the progression of NAFLD, lipid accumulation, and fibrosis.

In recent studies, BBR proved beneficial for ameliorating gut microbiota disruption and promoting, in this way, the establishment of specific bacteria [90]. In other studies, Dai Y. et al., Yang S. et al., and Chen D. et al. found that BBR can partially be involved in recovering intestinal integrity, thus exerting anti-inflammatory effects by targeting bacterial translocation [84,85,121]. In another study, Li H. et al. found that combining BBR and bicyclol improved gut microbiota and ameliorated lipid metabolism, thus exerting hepatoprotective properties [122].

#### 4.2.3. Sirtuins

Multiple studies in vivo and in vitro found different pathways through which SIRT3 may benefit NAFLD treatment. Xu X. et al. showed that mitochondrial β-oxidation was enhanced via the SIRT3 pathway after BBR treatment, proving that BBR exerts better lipid and glucose metabolism control. Moreover, in their study, Teodoro JS. et al. found that BBR completely restored mitochondrial activity; therefore, by upregulating SIRT3, there was a lower synthesis of ROS, and BBR proved in this way that it is an efficient anti-inflammatory agent. Zhang YP. et al. found similar results [97,98,110]. Another sirtuin on which recent studies showed that BBR activates is Sirtuin 1 (SIRT1). Wang P. et al. and Rafiei H. et al. proved that by upregulating SIRT1 expression, BBR managed to ameliorate changes induced by NAFLD, stimulated fatty acid oxidation, decreased intracellular lipid accumulation, as well as modulated mitochondrial respiratory subunits. Therefore, BBR via both SIRT1 and SIRT3 exerts significant lipid-lowering effects and anti-inflammatory properties [119,143].

#### 4.2.4. PPAR-Υ

Recent studies conducted on animal experimental models showed that BBR is an effective agonist of PPAR-Υ. In their research, Zhao W. et al. showed that BBR, in combination with other plant extracts used in traditional Chinese medicine, upregulated the expression of this receptor, thus exerting a significant anti-inflammatory effect as well as being able to alleviate histological changes, serum lipid profile, and liver enzyme levels. However, there were no significant changes compared to the Rosiglitazone treatment, which Zhao W. et al. used to correlate the outcomes with those obtained after BBR treatment [105]. Other studies, such as Ragab S.M. et al., further prove that BBR has lipid-lowering effects. BBR treatment significantly enhanced the expression of PPAR-Υ, thus ameliorating lipid metabolism in NAFLD-induced rats [108].

#### 4.2.5. Inflammatory Pathways

Inflammatory response due to high levels of pro-inflammatory cytokines as a result of liver exposure to gut-derived components is considered the main cause of the development and progression of NAFLD. If this inflammatory status is decreased, many changes associated with NAFLD might be improved. BBR was found to have multiple actions on various inflammatory pathways, thus proving that it can be a potent treatment for steatotic liver disease. These inflammatory pathways include mitochondria and oxidative stress, Nuclear Factor Kappa B (NF-κB), NRLP3, nuclear factor erythroid 2 related factor (Nrf2) pathway, and other modulating factors.

Mitochondria is the most important organelle involved in energetic metabolism. In addition to ATP synthesis and respiratory reactions, mitochondria are also responsible for cellular differentiation, proliferation, transduction, autophagy, and apoptosis [152]. Recent studies showed that BBR action on mitochondria proved to be mediated by various pathways. Xu X. et al. found that via SIRT3, mitochondrial β-oxidation was enhanced in BBR-treated mice. Therefore, a better control of lipid metabolism was obtained [98]. Moreover, Sun Y. et al. proved that BBR, via downregulation of Nrf2/HO-1 pathway expression, managed to decrease reactive oxygen species levels by particularly influencing I and III respiratory chains [130].

Regarding its anti-inflammatory properties, Teodoro JS et al. further demonstrated that by upregulating SIRT3 expression, BBR completely restored mitochondrial activity [110]. Similarly, but this time via the AMPK/SIRT1 pathway, Rafiei H. et al. found that mitochondrial activity was enhanced, and different proteins involved in lipid metabolism were phosphorylated, resulting in anti-steatotic, as well as anti-inflammatory effects in the cell culture model [143,144]. Dai Y. et al., Ye C. et al., and Deng Y. et al., in their animal and cellular experiments, further emphasize BBR’s antioxidant properties as shown by a decrease in pro-inflammatory cytokines levels, amelioration of liver histological levels and an improvement in oxidative stress response [100,115,123].

Nuclear factor kappa B (NF-κB), a transcription factor that participates in the inflammatory cascade associated with NAFLD, was found to be an essential target for BBR [153]. Zhao J. et al., in their study, proved that a combination of Metformin and BBR was superior for ameliorating steatosis and dyslipidemia in NAFLD-induced rats as BBR managed to inhibit TLR4/NF κB p65 signaling. By doing so, BBR managed to block the production of inflammatory cytokines [101]. Similar outcomes concerning inflammation were as well obtained by Wang L. et al. and Chen Y. et al. BBR modulated inflammatory response both in vivo and in vitro via TLR4/NF κB—lower levels of TNF-α, IL-6, IL-1β, amelioration of liver histology, proving in this way to be an effective hepatoprotective agent [120,140].

NLR family pyrin domain containing 3 is a protein complex that plays critical roles in the progression of steatotic liver disease and inflammatory response associated with NAFLD [126]. Bacterial or virus components, reactive oxygen species, and fatty acids can activate NRLP3 inflammasome, which in return stimulates interleukin-1b (IL-1b) and IL-18 synthesis [154]. Recent in vivo and in vitro studies found that BBR significantly inhibited NRLP3, thus balancing stress response and exerting important anti-inflammatory properties [124,126]. Moreover, in their research, Zhang Y. et al. found that Demethylene-tetrahydro berberine had superior effects to BBR or DMB alone in terms of their antioxidative properties [124].

Another mechanism through which BBR functions is the nuclear factor erythroid 2 related factor, a regulator of cellular response to oxidative stress [155]. BBR treatment was associated with decreased levels of genes involved in the inflammatory response. Moreover, by activating the Nrf2/ARE signaling pathway, BBR exerted anti-inflammatory effects by modulating different key genes associated with the oxidative stress response [100,130].

By inhibiting the mitogen-activated protein kinase (MAPK) pathway, BBR blocked the inflammation and angiogenesis associated with NAFLD in high-fat diet (HFD)-induced mice and showed potential anti-tumor effects, suggesting it might help prevent hepatocellular carcinoma [99]. In addition, He Q. et al. proved that BBR has a significant lipid-lowering effect both in vivo and in vitro, as shown by inducing liver autophagy and preventing liver lipid accumulation [134]. Similar effects were obtained after the co-administration of BBR and Sitagliptin by Mehrdoost S. et al. [88]. These further emphasize the beneficial properties of BBR against NAFLD and its complications [131,140].

#### 4.2.6. Other Pathways

Circadian clock proteins (clock and Bmall) are a group of proteins considered to be involved in controlling metabolic equilibrium [155]. Previous studies demonstrated that modifying the expression of these genes leads to the disruption of different molecular and metabolic processes, including glucose and lipid homeostasis, thus proving that these proteins take part in the pathophysiology of various diseases, including fatty liver disease [156]. BBR significantly alleviates Clock/Bmall1 oxidative stress, proving that it may play a protective role by rebalancing circadian rhythm and modulating lipidic, glucose metabolism, and inflammatory response associated with NAFLD [115]. Yang QH. et al. found that downregulation of mRNA of UCP2 after BBR treatment was associated with modulation of both inflammatory response and lipid metabolism in NAFLD-induced mice [111].

Adiponectin and angiopoietin-like protein 2 (ANGPTL2) are two proteins secreted by adipocytes, and both take part in inflammation. While large quantities of (ANGPTL2) are produced in response to inflammation, adiponectin ameliorates inflammatory response by modulating different signaling pathways [157,158]. BBR significantly downregulated the synthesis of pro-inflammatory proteins and cytokines by decreasing the expression of adipocyte macrophage-derived Angptl2 signaling pathway NAFLD-induced rats [95].

Neutrophil gelatinase-related lipoprotein (LCN2), a part of the lipocalin transport protein family, is mainly known for its role in inflammation. Recent studies showed that patients with NAFLD have higher levels of serum LCN2, thus proving to be a potential blood marker for early detection of NAFLD [159]. He H. et al. found that 8-cetyl berberine (CBBR) not only managed to modulate different genes involved in lipid accumulation, inflammation, or steatosis, but it also downregulated the LCN2 pathway relieving metabolic changes associated with NAFLD, showing in this way that maybe BBR and its derivatives might target inflammation and progression of steatotic liver disease [117]. In recent studies conducted both on animal experiment models and on cell cultures, BBR was able to inhibit free fatty acid and inflammatory response induced by LPS presence. In addition, BBR was shown to decrease TNF-α, IL-6, IL-1β, and MCP-1 production, proving to have significant anti-inflammatory effects [140].

Other pathways that were found to be influenced by BBR treatment both in vivo and in vitro are chemerin/CMKLR1 signaling pathway, SIRT1-FoxO1-SREBP2 pathway, insulin receptor substrate-2 (ISR-2), FXR/FGF-15, L-pyruvate kinase (LK), phosphatidate-phosphohydrolase activity, SETD2 pathway, and histones activity expression of different genes involved in lipid metabolism CPT-1, MTTP, and LDLR. Similar results were found concerning improvement in glucose and lipid metabolism, amelioration of histological changes induced by HFD, modulation of gut microbiota, alleviation of liver inflammation, and decrease of insulin resistance [87,95,109,112,113,123,136,139].

Regarding other secondary outcomes, BBR was most of the time associated with improvement of liver enzyme profile, serum lipid profile, body weight, amelioration of histological NAFLD-induced changes, restoration of lipid and glucose homeostasis, as well as a better control on stress oxidative response as shown by a decrease in pro-inflammatory cytokine levels. These all suggest that BBR exerts important hepatoprotective and anti-inflammatory properties, opening new directions for future NAFLD treatments. Recent studies, both clinical and experimental models, that we included in our systematic review proved that this natural compound not only modulates lipid metabolism but also targets mitochondrial activity, inflammatory cascade, and bacterial translocation. Consequently, BBR acts on the main pathophysiological mechanisms involved in NAFLD.

## 5. Conclusions

The strength and novelty of our review lie in its comprehensive approach. It is a systematic review covering both clinical and pre-clinical studies, thoroughly examining Berberine’s effects on NAFLD. Most of the literature we found highlights the beneficial effects of BBR on metabolic and molecular reactions involved in both lipid and glucose metabolism. Moreover, recent studies we researched in our review proved that this natural compound not only modulates lipid metabolism but also targets mitochondrial activity, inflammatory cascade, and bacterial translocation. By doing so, BBR acts on the main pathophysiological mechanisms involved in NAFLD.

In the clinical trials on human subjects diagnosed with NAFL/NASH, the quantifying methods showed a relevant reduction of the liver fat content and improved serum lipids, blood glucose levels, and liver enzymes associated with weight loss. In experimental models, Berberine was involved in the significant pathways preventing liver steatosis’s evolution. Although the available drugs are scarce or under development or under release, BBR seems to offer an attractive profile, which needs formulation improvement to facilitate the bioavailability of this natural compound.

## Figures and Tables

**Figure 1 ijms-25-04201-f001:**
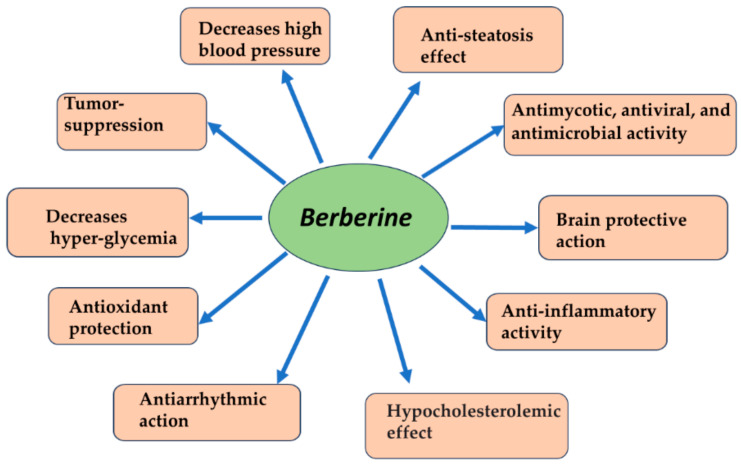
Multiple effects of Berberine.

**Figure 2 ijms-25-04201-f002:**
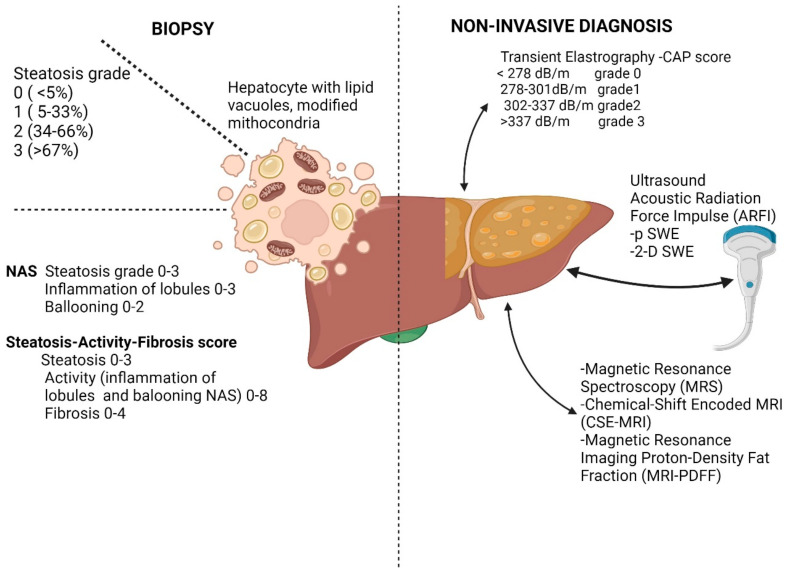
Non-invasive methods and histology criteria in NAFLD diagnosis and staging [13,14]. NAS—NAFLD Activity Score, CAP—controlled attenuation parameter, SWE—Shear Wave Elastography, MRI—Magnetic Resonance Imaging (Created with Biorender.com).

**Figure 3 ijms-25-04201-f003:**
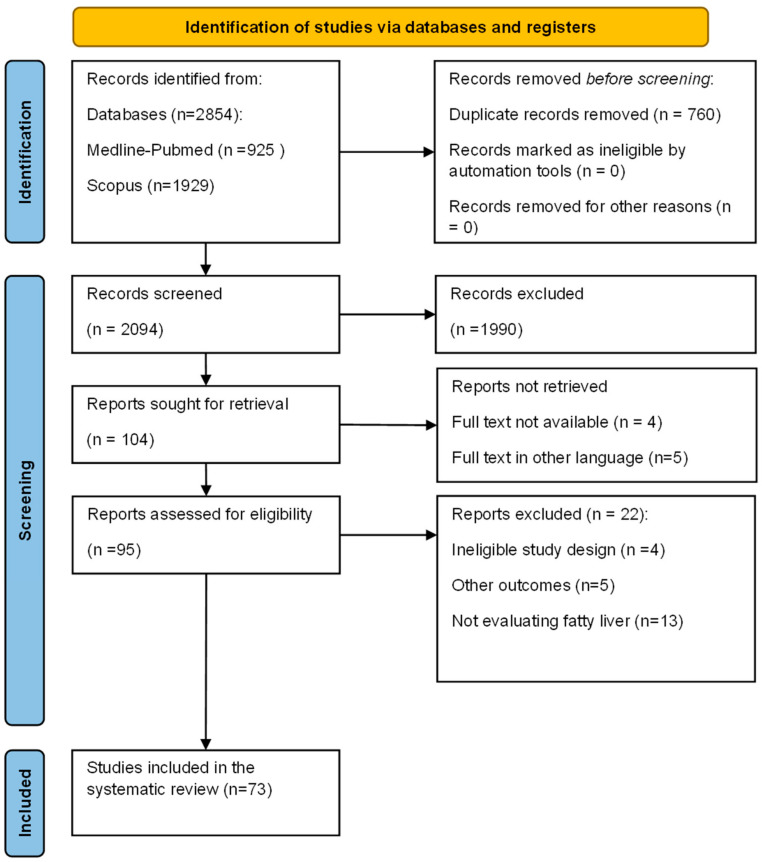
Flow Diagram for the study: Preferred Reporting Items for Systematic Reviews and Meta-Analyses (PRISMA).

**Figure 4 ijms-25-04201-f004:**
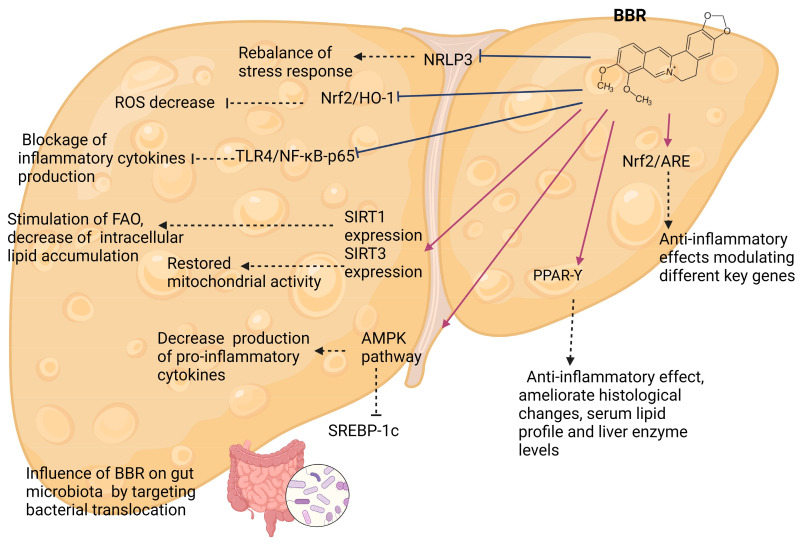
The main Berberine modulation pathways in NAFLD (Created with Biorender.com). NRLP3—NACHT, LRR, and PYD domains-containing protein 3 inflammasome, ROS—reactive oxygen species, Nrf/HO-1-Nrf2—nuclear factor erythroid-related factor 2/Heme Oxygenase 1-related factor, FAO—fatty acids oxidation, Nrf/ARE—nuclear factor erythroid-related factor/antioxidant response element, SIRT1—Sirtuin 1, SIRT3—Sirtuin 3, TLR4/NF-kB-p65—toll-like receptor 4/nuclear factor kB-p65, AMPK—Adenosine Monophosphate-Activated Protein Kinase, SREBP-1c—sterol regulatory element-binding protein 1c, BBR—berberine, PPAR-γ—peroxisome proliferator-activated receptor-γ (→ activation mechanism, ˫ downregulation, --- effect).

**Table 1 ijms-25-04201-t001:** Clinical studies with berberine as an interventional agent in NAFLD.

Authors	Clinical Study Type	Pharmaceutical Intervention and Dosage	Study Participants	Results and Key Impacts on NAFLD	NAFLD Assessments Methods
Harrison SA. et al., 2021 [68]	Randomized controlled, double-blind, placebo Phase 2	Berberine ursodeoxycholate (BUDCA) (18 weeks) BUDCA 500 mg bid/BUDCA 1000 mg bid/ placebo.	88 (27/30/32)	MRI-PDFF liver fat content was 2.4 times lower in the BUDCA higher dose group compared to placebo (4.8% vs. 2.0%) Secondary endpoints: Improvement in liver enzyme profile—mean ALT levels were 6.3 times lower than in the placebo group, and GGT levels were 15 times lower than in the placebo group. Regarding the serum lipide profile, BUDCA in higher doses shows better results in lowering LDL-c levels.	MRI-PDFF, Liver enzyme profile and serum lipid profile
Yan HM. et al., 2015 [69]	Randomized, parallel, multicenter, controlled, open-label clinical trial.	BBR 500 mg tid p.o 30 min before meal (16 weeks) Group A-lifestyle intervention (LSI)/Group B-LSI plus PGZ (15 mg q.d.) pioglitazone/Group C-LSI plus BBR	155 (53/47/55)	^1^H-MRS hepatic fat content was shown to improve better when the subject received lifestyle intervention and BBR rather than just dietary changes. (−17.4% vs. −12.1%, *p* < 0.008). Secondary endpoints: Improvement in liver enzyme profile as well as in serum lipide profile.	Hydrogen-1 MR spectroscopy (^1^H MRS) Liver enzyme profile and serum lipid profile
Ruiz-Herrera VV. et al., 2023 [70]	Randomized, double-blind, placebo-controlled clinical trial	BBR 500 mg tid (20 weeks) BBR tid/placebo (calcinate magnesia) 400 mg tid	36 (19/17)	The BBR-treated group showed slight amelioration regarding lipid profile, but this was not statistically relevant. Secondary endpoints: Better control of glucose metabolism. Relevant improvements regarding Metabolic Syndrome features were found in the current study.	Liver enzymes profile and serum lipid profile.
Chang X. et al., 2016 [71]	Randomized, parallel-controlled, open-label, clinical trial	BBR 500 mg tid 30 min before meals (16 weeks) Group A-LSI/Group B-LSI plus BBR	41 BBR/39 LSI 80 PTS	^1^HMRS liver fat content decreased in the BBR-treated group compared with the LIS-only group. (13.6% vs. 20.3%, *p* = 0.021) Secondary endpoints: The lipid-lowering effect of BBR was reflected by various changes in lipid profile.	Hydrogen-1 MR spectroscopy (^1^H MRS) Liver enzyme profile and serum lipid profile
Geng Q. et al., 2022 [72]	Retrospective cohort	BBR 500 mg tid (16 weeks) BBR + Dietary modification and exercise/Bicyclol + Dietary modification and exercise /Dietary modification and exercise	385 (112/145/128)	The Liver/spleen CT ratio was significantly improved in the BBR-treated group with a *p*-value < 0.0001. Liver steatosis was notably attenuated on liver biopsies after BBR treatment (4.44 vs. 5.28, *p* < 0.0001), as shown by an improved NAS score. Secondary endpoint: BBR treatment ameliorated liver function tests, as shown by a significant decrease in both AST and ALT levels (*p* < 0.0001, respectively, *p* < 0.001). No significant improvement in serum lipid profile was recorded in the BBR-treated group.	Liver/spleen ratio CT scans NAFLD activity score on liver biopsy (NAS score) Liver enzyme profile and serum lipid profile
Nejati L. et al., 2022 [73]	Open-label, double-blinded, randomized, controlled trial	6.25 g daily p.o (7 weeks/45-day)	48 (24/24)	BBR treatment did not improve liver function tests.	Ultrasonography Liver enzymes profile and serum lipid profile.
Yan H. et al. (2021) [74]	Randomized, parallel controlled, open-label clinical trial with three-arm.	BBR 500 mg tid (16 weeks) (1) LSI (2) LSI + PZG (3) LSI + BBR	155 (85/70)	Liver fat content (LFC) was significantly decreased in BBR-treated women compared to lifestyle intervention alone (−11.88%, *p* = 0.02 BBR + LSI vs. LSI), and there were no significant outcomes in men. However, men responded better regarding LFC decrease in BBR + LSI vs. LSI + PZG (−11.29%, *p* = 0.07). BBR did not have a significant effect in reducing liver fat content compared to PZG or LSI groups (BBR + LSI vs. LSI, *p* = 0.124 and BBR + LSI vs. PGZ + LSI, *p* = 0.222). Secondary outcomes: Pioglitazone had superior effects on both glucose metabolism and liver fat content among women.	Ultrasonography. Hydrogen-1 MR spectroscopy (^1^H MRS).
Cossiga V. et al. (2019) [75]	Randomized controlled trial	Plant extracts consisting of: 500 mg BBR 30 mg tocotrienols 30 mg chlorogenic acid (duration not mentioned) (1) placebo (2) treated group	49 (26/23)	Significant improvement in median CAP value after treatment compared to placebo (*p* < 0.01). Secondary outcomes: At the end of the study, the liver enzyme profile and serum lipid profile did not significantly change. Better control of glucose metabolism in the treated group.	Transient elastography—CAP value. Liver enzymes profile and serum lipid profile.
Wu L., 2019 [76]	Interventional prospective study /pre-clinical study	BBR 500 mg tid, p.o, 30 min before meals (4 weeks)	10	^18^F-FDG PET/CT showed that 1 month of BBR treatment was associated with significantly enhanced recruitment and activity of brown adipose tissue (*p* < 0.05). Secondary endpoints: Mild improvement in basal metabolism after BBR treatment. No significant changes were found in the serum lipid profile.	^18^F-FDG PET/CT and Micro ^18^F-FDG PET/CT imaging (for mice). Liver enzymes profile and serum lipid profile.

^1^HMRS—Hydrogen-1 MR spectroscopy, ALT—alanine transaminase, AST—aspartate aminotransferase, bid—twice a day, BUDCA—berberine ursodeoxycholate, CT—computed tomography, GGT—gamma-glutamyl transpeptidase, LDL-c—low-density lipoprotein cholesterol, LSI—lifestyle intervention, MRI-PDFF—Magnetic Resonance Imaging Proton Density Fat Fraction, NAS score—NAFLD activity score, p.o—by mouth, PZG—pioglitazone, q.d—once a day, tid—three times a day, ^18^F-FDG—[^18^F]2-fluoro-2-deoxy-D-glucose, PET/CT—Positron Emission Tomography-Computed tomography scan.

**Table 2 ijms-25-04201-t002:** In vivo pre-clinical studies of Berberine’s effects on NAFLD pathways.

Authors	Experiment and Model Type	Pathway
Chen D. et al. (2023) [84]	Sprague–Dawley rats	Liver and spleen microbiota cultures. Genomic DNA analysis for gut microbiota.
Dai Y. et al. (2022) [85]	Sprague–Dawley rats	Genomic DNA analysis for gut microbiota. ELISA for inflammatory cytokines levels.
Wang Y. et al. (2021) [86]	Mixed background C57Bl/6J and 129S1/SvlmJ (B6/12) mice	Increased expression and analysis of multiple cellular pathways involved in NAFLD induction and progression.
Shu X. et al. (2021) [87]	Mice C57BL/6J	Upregulation of FXR pathway and FGF-15 levels. Genomic DNA analysis for gut microbiota.
Mehrdoost S. et al. (2021) [88]	Sprague–Dawley rats	Upregulation of Adiponectin receptor 2 (AdipoR2) levels and mitogen-activated protein kinase (MAPK) ERK expression and their influence on inflammation and progression of NAFLD.
Li QP. et al. (2021) [89]	Sprague–Dawley rats	Activation of AMPK pathway
Cossiga V. et al. (2021) [90]	C57BL/6J mice (in vivo)	Genomic DNA analysis for gut microbiota.
Wang L. et al. (2021) [91]	Sprague–Dawley rats	ELISA for inflammatory cytokine levels Downregulation of nuclear translocation of NF-κB via the TLR4/MyD88/NF-κB pathway.
Lu Z. et al. (2021) [92]	Wild-type (WT) Wistar rats	Downregulation of chemerin/CMKLR1 signaling pathway and Treg/TH17 ratio.
Yu M. et al. (2021) [93]	C57BL/6J	Upregulation of both mitochondrial activity and AMPK pathway, Genomic DNA analysis for gut microbiota
Chen P. et al. (2021) [94]	Sprague–Dawley rats	Downregulation of triglyceride transfer protein (MTTP), apolipoprotein B, and low-density lipoprotein receptor (LDLR)
Lu Z. et al. (2020) [95]	Sprague–Dawley rats	Upregulation of adipocyte macrophage-derived Angptl2 signaling pathway analysis.
Wang W. et al. (2020) [96]	Sprague–Dawley rats	Upregulation of NOD1, NOD2 pathway. NLRP3 inflammasome activity
Zhang YP. et al. (2019) [97]	Rats Sprague–Dawley	Activation of SIRT3/AMPK/ACC pathway.
Xu X. et al. (2019) [98]	C57BL/6J mice	Activation of SIRT3 pathway and its effects on lipid metabolism. Stimulation of mitochondrial β-oxidation levels.
Wu L. et al. (2019) [76]	C57BL/6J mice	Micro ^18^F-FDG PET/CT. Temperature and metabolic activity measurements. Fat adipose tissue analysis.
Luo Y. et al. (2019) [99]	C57BL/6J mice	Downregulation of p38MAPK/ERK-COX2 pathway. Analysis.
Deng Y. et al. (2019) [100]	Sprague–Dawley rats	Upregulation of Nrf2/ARE signaling pathway and genes involved in the inflammatory response.
Zhao J. et al. (2018) [101]	Sprague–Dawley Rats	Activation of AMPK pathway.
Feng WW. et al. (2018) [102]	Sprague–Dawley rats	Downregulation of sterol regulatory element-binding protein 1c (SREBP-1)c, pERK, TNF-α, and pJNK-genes involved in inflammatory response.
Zhao L. et al. (2017) [103]	Sprague–Dawley rats	Liver enzyme profile and serum lipid profile. Histological analysis.
Yang J. et al. (2017) [104]	C57BL/6J Apolipoprotein E-deficient (ApoE-/-)	Activation of C-X-C chemokine receptor type 4 (CXCR4)/CXCL12 signaling pathway.
Zhao W. et al. (2016) [105]	Sprague-Dawley rats	ELISA for detecting pro-inflammatory cytokine levels (TNF-α and IL-6). Upregulation of PPAR-γ and Insulin receptor (IR) activation.
Cao Y. et al. (2016) [106]	Mice BALB/c	Liver enzyme profile and lipid profile. Histological analysis. RT-PCR for key enzymes and proteins.
Xue M. et.al. (2015) [107]	db/db mice	Downregulation of lipogenic genes: fatty acid synthase (FAS), stearoyl-CoA desaturase (SCD1), and sterol regulatory element-binding protein 1c (SREBP1c). Upregulation of lipolytic gene carnitine palmitoyltransferase-1 (CPT1).
Ragab SM. et al. (2015) [108]	Wistar rats	Upregulation of peroxisome proliferator-activated receptor γ (PPARγ) in adipose tissue and liver.
Heidarian E. et al. (2014) [109]	Wistar rats	Downregulation of phosphatidate phosphohydrolase (PAP) activity. Antioxidants level assay.
Teodoro JS. et al., (2013) [110]	Sprague–Dawley-rats	Stimulation of mitochondrial activity and reactive oxygen species levels. Upregulation of SIRT3 activity level measurements.
Yang QH. et al. (2011) [111]	Sprague-Dawley rats	Downregulation of uncoupling protein-2 (UCP2) mechanism.
Xing LJ. et al. (2011) [112]	Wistar rats	Significant upregulation of insulin receptor substrate-2 (ISR-2) levels analysis.
Chang XX. et al. (2010) [113]	Sprague-Dawley rats /buffalo rat liver (BRL)	Downregulation of genes associated with lipid metabolism: CPT-1, MTTP, and LDLR.

**Table 3 ijms-25-04201-t003:** Mixed studies (in vivo and in vitro) regarding the effects of Berberine in NAFLD pathways.

Authors	Experiment and Model Type	Pathway
Ma X. et al. (2024) [114]	HepG2 cell culture/BALB/c mice	Upregulation of various genes involved in lipid metabolism.
Ye C. et al. (2023) [115]	C57BL/6J mice/HepG2 cell culture	ELISA for inflammatory cytokine levels. Downregulation of Clock and Bmall1 genes activity.
Guo HH. et al. (2023) [116]	ApoE(−/−) mice/Caco-2 cells	Genomic DNA analysis for gut microbiota. Downregulation of TLRs/NF-κB signaling pathway including TLR2, TLR4, and IKKβ. Upregulation of AMPK pathway, ELISA for inflammatory cytokine levels.
He H. et al. (2023) [117]	L02 and HepG2 cell lines C57BL/6*J* mice	Downregulation of lipocalin-2 (LCN2) activity.
Zhou LM. et al. (2023) [118]	C57BL/6 J wild-type (WT) mice and/HepG2 cell lines	Suppression of SREBP1/FASN pathway analysis.
Wang P. et al. (2022) [119]	HepG2 and AML12 cells line/C57BL/6J mice	Upregulation of SIRT1 and CPT1A activity.
Chen Y. et al. (2021) [120]	C57BL/6 J mice/Caco-2 human colon cancer cell line LO2 human normal liver cell line	Downregulation of NF-κB pathway activity.
Yang S. et al. (2022) [121]	Mice C57BL/6J HepG2 cell cultures	Liver enzyme profile and serum lipid profile. Genomic DNA analysis for gut microbiota.
Li H. et al. (2022) [122]	Mice C57BL/6J/HepG2 cells	Genomic DNA analysis for gut microbiota.
Dai L. et al. (2022) [123]	HepG2 cells culture/C57BL/6J mice	Downregulation of SETD2 pathway and histones activity (H3K36me3)
Zhang Y. et al. (2021) [124]	C57BL/6J mice/L02 cell culture	Pro-inflammatory cytokines levels analysis. Suppression of NRLP3 pathway and stress oxidative response.
Ma CY. et al. (2021) [125]	C57BL/6J mice, ApoE−/− mice/Hep G2 cells	Activation of MAPK/ERK1/2 signaling pathway. Downregulation of PCSK9 expression.
Mai W. et al. (2020) [126]	AML12 cell culture and/C57BLKS/J mice (in vivo and in vitro)	Suppression of Caspase-1 and Nod-like receptor family pyrin domain containing 3 (NLRP3) inflammasome activities analysis and ROS.
Zhu X. et al. (2019) [127]	C57BL/6 J and ob/ob mice/HepG2 and AML12 cells and six liver biopsies from NAFLD patients	Downregulation of AMPK-SREBP-1c-SCD1 pathways.
Sun Y. et al. (2018) [128]	C57BL/6J mice/hep G2 cell cultures	Upregulation of SIRT1 pathway/Autophagy/FGF21 activity.
Liang H. et al. (2018) [129]	C57BL/6J mice/Cell cultures	Downregulation of ATP-binding cassette transporter A1 (ABCA1) and protein kinase C δ pathway activities.
Sun Y. et al. (2017) [130]	C57BL/6J mice/Sprague Dawley rats/Human hepatoma cell lines Huh7 and HepG2	Mn-SOD activity measurement and mitochondrial respiratory chain activity. Downregulation of Nrf2/HO-1 pathway activity.
Choi YJ. et al. (2017) [131]	Mice C57BL/6J/cell cultures HepG2	Activation of AMPK, ERK-C/EBPβ pathways, and CD36 expression.
Zhang Z. et al. (2016) [132]	Mice C57BL/6J (db/db)/HepG2, FAO cell lines. AML 12	Activation of ATF6/SREBP-1c pathway.
Qiang X. et al. (2016) [133]	Mice C57BLKS/J (db/db)/ICR mice	Activation of AMPK pathway.
He Q. et al. (2016) [134]	LO2 human cell line/C57BL/6 mice	Enhanced autophagy induced by activated ERK-dependent mTOR pathway.
Guo T. et al. (2016) [135]	Mice C57BL/6J/H4IIE cells (rat hepatoma cells)	Histological analysis. Western blot and RT-PCR for key enzymes and proteins.
Zhang Y. et al. (2015) [136]	Wistar rats/Cell cultures	Upregulation of L-pyruvate kinase (LK) activity.
Yuan X. et al. (2015) [137]	Sprague-Dawley rats/Huh7 Human hepatic cell line	Downregulation of various IncRNAs and mRNAs associated with NAFLD.

**Table 4 ijms-25-04201-t004:** In vitro pre-clinical studies of Berberine’s effects on NAFLD pathways.

Authors	Experiment and Model Type	Pathway
Rafiei H. et al. (2023) [138]	Cell cultures HepG2,/LX-2 stellate cells, differentiated THP-1 cells	ELISA to quantify levels of inflammatory cytokines and chemokines. Lower levels of ROS.
Shan MY. et al. (2021) [139]	HepG2 cell culture	RT-PCR and Western blot for key enzymes. SIRT1-FoxO1-SREBP2 pathway downregulation.
Wang Y. et al. (2020) [140]	Macrophages (RAW264.7) and hepatocyte cell lines	Downregulation of endoplasmic reticulum stress response and ERK1/2 pathway. Suppression of PA/LPS-induced inflammation.
Sharma A. et al. (2020) [141]	Cell cultures HepG2 and Hepa 1–6	Activation of AMPK/mTOR/SREBP-1c and AMPK/Nrf2 modulate lipid metabolism and inflammatory and oxidative stress.
Babaei Khorzoughi R. et al. (2019) [142]	HepG2 cell cultures	Downregulation of SREBP-1c and FAS expressions
Rafiei H. et al. (2019) [143]	HepG2 cell cultures	Upregulation of both mitochondrial and AMPK/SIRT1 pathway activity.
Rafiei H. et al. (2017) [144]	HepG2 cell line	Fluorescence analysis for reactive oxygen species analysis. Effects of different polyphenols against stress oxidative response.
Liu Y. (2017) [145]	HepG2 cell cultures	Downregulation of FXR/SREBP-1c/FAS pathway.
Brusq JM. et al. (2006) [146]	HepG2 cell cultures	Activation of AMPK pathway.

^18^F-FDG PET/CT—positron emission tomography with 2-deoxy-2-[fluorine-18]fluoro-D-glucose integrated with computed tomography, ACC—acetyl-CoA, AdipoR2—adiponectin receptor 2, AML 12—immortalized mouse normal hepatocytes, AMPK—Adenosine Monophosphate-Activated Protein Kinase, ATF6—activating transcription factor 6, ATP—adenosine triphosphate, BALB/c—albino bred mice, Bmall—brain and muscle aryl hydrocarbon receptor translocator-like protein 1, C—total cholesterol, C57BL/6J—pathogen-free mice, Caco2—human colon cancer line, CD36—a cluster of differentiation 36, CMKLR1—chemokine-like receptor 1, CPT1—carnitine palmitoyltransferase-1, DNA—deoxyribonucleic acid, EBPβ—CCAAT/enhancer-binding protein beta (C/EBPβ), ELISA—enzyme-linked immunosorbent assay, ERK—extracellular regulated-signal kinase, FAO—rat hepatoma line, FAS—fatty acid synthase, FGF-15—fibroblast growth factor 5, FGF21—fibroblast growth factor 21, FoxO1—forkhead-box transcription factor 1, CCAAT/enhancer-binding protein beta (C/EBPβ), FXR—farnesoid x receptor, H4IIE—rat hepatoma cell line, HDL-c—high-density lipoprotein, HepG2—human hepatoma line, HO-1—heme oxygenase-1, Ig—intragastrical administration, IKKβ—inhibitor of κB kinase, IL-1β—interleukin 1-β, IL-6—interleukin 6, IRT1—iron-regulated transporter 1, LDLR—low-density lipoprotein receptor, LO2—human normal liver cell line, LPS—lipopolysaccharide, MAPK—mitogen-activated protein kinase, mTOR—mammalian target of rapamycin, MTTP—triglyceride transfer protein, MyD88—myeloid differentiation primary response protein 88, NF-κB—nuclear factor kappa-light-chain enhancer of activated B cells, NLRP3—nod like receptor protein 3, NOD1—nucleotide-binding oligomerization domain-containing protein 1, NOD2—nucleotide-binding oligomerization domain-containing protein 2, Nrf2—nuclear factor erythroid 2-related factor, PAP—phosphatidate phosphohydrolase, PA—palmitic acid, PCSK9—proprotein convertase subtilisin/kexin type 9, pERK—protein kinase R-like ER kinase, ROS—reactive oxygen species, RT-PCR—reverse transcription polymerase chain reaction, SCD1—stearoyl-CoA desaturase, SIRT3—mitochondrial NAD-dependent deacetylase sirtuin-3, SOD—superoxide dismutase, SREBP-1c—sterol regulatory element-binding protein 1, SREBP2—sterol regulatory element-binding protein 2, TLR2—toll-like receptor 2, TLR4—toll-like receptor 4, TNF-α—tumor necrosis factor-α, Wistar—albino bred mice.

## Data Availability

Data is contained within the article and Appendix A.

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
