# Peer review of "Berberine Effects in Pre-Fibrotic Stages of Non-Alcoholic Fatty Liver Disease—Clinical and Pre-Clinical Overview and Systematic Review of the Literature"

_ijms, 2024, doi:10.3390/ijms25084201_

Round 1

Reviewer 1 Report

Comments and Suggestions for Authors

The authors were trying to review the effects of berberine on NAFLD.  However, there are some suggestions are listed as followed:

1.Line 59-61,the information is repeated as line 55-58.

2.Line 62-65 and line 78-79,the information is similar.

3.Line 86-87,in same sentence,same paper was cited twice.

4.Line 83-87,similar descriptions for feature of MAFLD,which has been mentioned previously.

5.There is no any valuable information about diagnosis in section “1.1.1. Pre-Fibrotic Stages in NAFLD”. This section can be discarded.

6. Serum markers/biochemical parameters such as ALT,AST,TG etc. are very useful and important for liver diseases including MAFLD, more information is required in this section “1.1.4. Serum Markers in NAFLD”

7. There is no clear standard discussed for grade/stage classification in section “1.1. Diagnosis of Liver Steatosis” and this section is suggested to be categorized by diagnosis methods basing on the current manuscript.

8.Berberine is the main topic but there is no any information for berberine in the introduction.

9.The authors presented figure 2 and then figure 1 .

10. Line 272, there is no full name for BBR when it showed for the first time.

11. Line 278-280, the legend is not complete and not well recognized as legend.

12. Conclusion or discussion is required for “section 3.1. Clinical Studies on BBR and NAFLD”.

13. 13.The title of table 2 and 4 should be described completely so that people can understand the content in table from the title.

14.How were the pathways or molecules changed in table 2-4 ï¼Ÿ Up-regulated or down-regulated, activated or inhibited ? Please mark.

15.There is no a clear or specific conclusion for mechanism of berberine effect on NAFLD in the results, or what’s the main mechanism behind ? what’s the regulation network?

16.The results part looks like a paper list of berberine in vivo and in vitro work. The content should be enriched.

17. In section “Pathophysiology and Molecular Signaling”, what’s the regulation network for those molecules mentioned in this section?

18.Some content about berberine found from the analysis is suggested to be included in the results rather than discussion part.

Comments on the Quality of English Language

Minor editing of English language required.

Author Response

Dear Reviewer,

We want to thank you for taking the time to assess our manuscript and appreciate the effort you dedicated to providing feedback on it. We appreciate your comments, which will help improve our manuscript. We carefully considered your suggestions, addressed every one of them, and highlighted them in the manuscript.

1 .Line 59-61, the information is repeated as line 55-58-

Response: We corrected the text.

  1. Line 62-65 and line 78-79,the information is similar-

Response: We rewrote the content.

  1. Line 86-87,in same sentence,same paper was cited twice.

Response: We corrected

  1. Line 83-87,similar descriptions for feature of MAFLD,which has been mentioned previously.

Response: Although that might be similar, we modified the content to be more clear.

  1. There is no any valuable information about diagnosis in section “1.1.1. Pre-Fibrotic Stages in NAFLD”. This section can be discarded.

Response: We discarded this section

  1. Serum markers/biochemical parameters such as ALT,AST,TG etc. are very useful and important for liver diseases including MAFLD, more information is required in this section “1.1.4. Serum Markers in NAFLD”

Response: We enriched our content about Serum Markers in NAFLD and highlighted this in the manuscript.

‘For the time being, liver biopsy is considered to be the gold standard when it comes to NAFLD accurate diagnosis. However, carrying out biopsies every time steatotic liver disease is suspected remains controversial, and that is why tedious research has been directed towards less invasive tests that can offer a better alternative [38]. These new research areas consist of different markers and scoring systems that could potentially replace invasive or expensive methods in the near future. While available tests for the fibrotic stages may be helpful in clinical practice, the serum panels or biomarkers need further validation for NASH. Regarding apoptosis markers cytokeratin (CK) -18, a hepatocyte apoptosis fragment was the most studied for NAFLD diagnosis [39]. Even though CK -18 is unavailable for usual practice and more feasible for trials [18], [29], recent studies showed a promising result regarding NAFLD diagnosis especially in combination with FIB-4 (fibrosis 4 index) test and MACK-3 test (a new blood test consisting of Homeostatic Model Assessment- Index Insulin Resistance (HOMA-IR), aspartate aminotransferase (AST) and CK18 levels) [40]

Furthermore, the perspective of gene-based markers seems attractive. Similarly to CK18, the clinical utility is feeble for the NASH score that relies on the Patatin-like phospholipase domain-containing protein 3 (PNPLA3) genotype or tests for microRNA profiling [29].  Zeng Y.. et al. 2022 proved in their study that the so-called omics biomarkers PNPLA3, TM6SF2 (transmembrane 6 superfamily 2 human gene), different types of microRNAs such as miRNA-122, miRNA-199, miRNA-34a or extracellular vesicles could open new doors to future diagnostic methods for steatotic liver disease. Still, these need further studies [39].

In addition to all of these, there are several tests proposed to diagnose NAFLD, the six most studied being involving the fatty liver index (FLI), AST, platelet ratio index (APRI), FIB-4 index, AST/ALT 4 ratio, Bard score, and NAFLD fibrosis score (NFS) [41]. Most of these co-scoring tests consist of a combination of serum biomarkers and anthropomorphic parameters such as age, AST, gamma-glutamyl transferase (GGT) levels, platelet count, serum albumin levels, international normalized ratio (INR), impaired fasting glycemia or presence of diabetes body mass index [42], [41]. Although they were demonstrated to be effective in excluding severe cases of fibrosis, none of them have proven to differentiate between the lower and more advanced stages of liver fibrosis [43]. Another promising panel blood base biomarker appears to be NIS-4, which includes markers representative of NASH, such as α2-macroglobulin, miR-34a-5p, YKL-40, and hemoglobin A1c (HbA1c). In the study of Harrison et al., this marker was not influenced by age, weight, or liver enzymes. This suggests that it could be a valuable tool in identifying features of NASH [32].

No established test, marker, or algorithm can accurately diagnose NAFLD; recent research has found new and innovative ways through which we hopefully will replace liver biopsy as the sole precise method for steatosis liver disease diagnosis.”

  1. There is no clear standard discussed for grade/stage classification in section “1.1. Diagnosis of Liver Steatosis” and this section is suggested to be categorized by diagnosis methods basing on the current manuscript.

Response: We reorganized this chapter to be more comprehensible.

8.Berberine is the main topic but there is no any information for berberine in the introduction.

Response: As you well suggested the topic about Berberine is written in the introduction section

“Berberine represents one of the plant's secondary compounds: Benzyl-iso-quinoline alkaloids (BIAs). These natural alkaloid compounds are found in the outermost layer of stems and roots of multiple therapeutic plant species from the Berberidaceae family, genus Berberis. [10]

Figure 1. Multiple effects of Berberine

The interest in Berberine has increased in recent years due to its multiple benefits, as botanical alkaloids such as Isoquinoline alkaloids have been proven to have versatile medical effects. As poly-pharmacological compounds, alkaloids showed antiphlogistic, antiseptic, antioxidant, pain-relieving, carcinogenetic-inhibiting, antispasmodic, and anti-tussive effects.[10],[11],[12] (Figure     1)

Berberine and its metabolites and derivatives have various medical actions. They have a multiorgan distribution, with the utmost concentration at the hepatic level. However, they also have a high action in the renal, muscles, respiratory, cerebral, heart, and pancreas and the least amount in fatty tissue [10].”

9.The authors presented figure 2 and then figure 1 .

Response: We corrected the numeration of our tables and figures.

  1. Line 272, there is no full name for BBR when it showed for the first time.

Response: We corrected the abbreviations throughout the manuscript.

  1. Line 278-280, the legend is not complete and not well recognized as legend.

Response: We wrote the legends of tables and figures in italics to better acknowledge them.

  1. Conclusion or discussion is required for “section 3.1. Clinical Studies on BBR and NAFLD”.

Response: We have a comprehensive discussion for section 3.1 in section 4. Discussion.

“In their phase 2 study, Harrison et al. showed that BUDCA treatment might be an option for treating patients with NAFLD and other characteristics of metabolic syndrome. Not only did they find a significant reduction in liver fat content after high-dose BUDCA treatment, but this combination of berberine and ursodeoxycholate also significantly impacted glucose profile. Glycemic control was better controlled by higher dose BUDCA treatment, as seen by the reduction in HbA1c mean levels in this group (two times more than in low-dose BUDCA treatment, 0.6% vs 0.3%). No changes in fasting blood glucose, HOMA-IR, and insulin levels were detected. BUDCA high-dose treatment was shown to have a significant impact on weight loss. Subjects lost more than twice the weight lost by those on lower dose BUDCA treatment (-3.5% vs -1.6%, p=0.012). Even though BUDCA had a significant impact on liver function, surprisingly, there was no change in HDL-c, and TG levels were better controlled by lower doses of BUDCA (a decrease by 41 mg/dl in this group vs 24 mg/dl in 1000 mg BUDCA treatment). Also, the authors stated that the formula they used in this study, HTD1801 (berberine ursodeoxycholate or BUDCA), distinguishes itself from these agents in both structure and function and is different from other investigational agents used in NAFLD treatment, such as obeticholic acid, functioning as a farnesoid X receptor (FXR) agonist bile acid, aramchol, a liver-targeted SCD-1 inhibitor, and the thyroid hormone β agonist- resmetirom. [68], [149].

Another factor in the NAFLD pathogenesis pathway is FGF-19, which acts as a hormonal regulator. However, as shown by Harrison et al., BUDCA treatment had no significant impact on serum FGF-19 or fibrotic indicators such as propeptide type III collagen (Pro-C3) and Enhanced Liver Fibrosis (ELF) score [68].  

In the study of Yan et al., although the BBR group had similar changes in hepatic fat content with the pioglitazone group (the positive control reference for this study), the berberine cohort exhibited notably greater reductions in liver fat content compared to those in the LSI group, alongside more pronounced decreases in blood glucose levels, triglycerides, and cholesterol. Moreover, similar results concerning glucose homeostasis and liver enzyme levels were discovered between the BBR-treated group and the Pioglitazone group, but BBR was more efficient in reducing body weight and improving serum lipid profile [69].

To evaluate the safety and bioavailability of BBR from the LIS alone and BBR-treated groups, eleven subjects were randomly selected whose blood and urine were further tested. The results showed that BBR  presented good absorption, and regarding the serum BBR and its metabolites, concentration was 50 times lower compared with the hepatic uptake. This suggests that BBR may activate different hepatic genes that are involved in lipidic and glucose metabolism[69].

In the study of Herrera et al., the results were relevant, especially for the changes in the body mass index, which were greater in the BBR-treated group, and lower serum levels of triglycerides, total cholesterol, and LDL-c, alongside a better glycemic profile in these patients. There were no significant changes in ALT or AST after 20 weeks of BBR treatment [70]. Interestingly, the study reported impact of BBR in tumor necrosis factor-α (TNF-α) and high sensitive C-reactive protein (hs-CRP)[70].

In the study by Nejati et al., it was reported that not only serum triglyceride (TG) levels were not improved after treatment with BBR, but also TG levels were reduced to a greater extent in the control group compared to the BBR-treated group. Furthermore, no significant decrease was observed in either group's total cholesterol (TC) levels or TG levels (p=0.326 and p=0.464, respectively). In addition, BBR proved not to significantly impact liver enzyme profile, fasting blood glucose, body weight, and BMI among subjects. The differences between the other studies with BBR as an interventional study agent is the formulation and dosage of BBR in this study, as the patients received 6.25 g daily prepared by boiling 100 g of dried BBR in 5000 ml of water until reduced to 4000 ml through boiling [73].  Since the bioavailability of BBR is dependent on the formulation, this preparation method may influence the concentration and absorption of BBR and explain the study's results. 

The research conducted by Chang et al. demonstrated that BBR lowered liver fat content, as quantified by 1HMRS, by an additional 7% compared to cases that only underwent lifestyle changes. There were no statistically relevant differences regarding liver enzyme levels in both groups [71]. BBR caused a more significant reduction in HFC than LSI alone. Furthermore, BBR treatment was associated with significant improvement concerning many areas: ameliorated anthropomorphic features (lower body weight, smaller waist circumference, BMI - p<0.01), decrease of liver fat burden - p<0.01; improved glucose homeostasis ( lower blood glucose, better control on glucose metabolism - HbA1c - p<0.01), and better control on lipidic metabolism (lower serum levels of cholesterol, triglycerides, LDL-c, apoA/B, LP(a) – p<0.05). In their study, Chang et al. observed that BBR – through its ability to modulate different types of lipids, had similar effects, with just lifestyle changes, on lipid metabolism, except the ceramides, which were influenced solely by BBR treatment [71].  As ceramides are involved in the NAFLD pathogenic pathways, their inhibition may prevent the evolution of NASH[71].

Insulin sensitivity was improved by decreasing the visceral and subcutaneous adipose tissue after BBR treatment. In addition to this, HOMA-IR was also improved in subjects that underwent BBR treatment.

To further test these findings among humans, the authors proposed an animal model using diet-induced obese mice. BBR treatment was also associated with changes in brown adipose tissue and thermogenesis and increased energy expenditure, as shown on 18F-FDG PET/CT. Furthermore, there was also an improvement in glucose metabolism in the animal model group.  [76]

In their study, Cossiga V. et al. (2019) tried to evaluate the effects of plant extracts consisting of Berberis aristata, Elaeis guineensis, and Coffea canephora on NAFLD patients. At the end of their experiment, they assessed the liver fat content by measuring mean CAP, and they found that treatment with these plant extracts significantly decreased mean CAP from 291.6 ± 39.2 dB/m to 251.3 ± 41.5 dB/m with a p-value < 0.01. This further proves that BBR could be an effective treatment for fatty liver disease. Moreover, there was a significant improvement in glucose metabolism in the treated group compared to the placebo group, as shown by lower levels of blood glucose levels (p<0.001), an improvement in HOMA-IR index (p<0.001), as well as lower levels of insulin (p<0.01). They found no significant changes in the liver enzyme and serum lipid profiles. Overall, their study by Cossiga V. et al. showed that BBR can potentially exert a hepatoprotective effect and improve glucose homeostasis associated with NAFLD.

Another study conducted on patients with NAFLD treated with BBR by Yan H. et al. 2021 studied the effects of pioglitazone, BBR, and lifestyle changes in men and women. Their results showed that pioglitazone intervention was superior for improving both lipid and glucose metabolism, mainly in women; regarding liver fat content in BBR-treated groups, women showed a decreased liver fat content when compared to the lifestyle-only group (p=0.020), while men have a better response in BBR group compared to pioglitazone treated group (p=0.007). Moreover, BBR was shown to have no significant interaction regarding efficacity and gender among the studied groups.

Regarding the side effects, BBR proved to be relatively safe. In all human studies, patients were regularly evaluated for safety reasons, and one of the frequent adverse reactions reported by the subjects was diarrhea. Others complained of nausea, anorexia, dyspepsia, abdominal pain or distention, constipation, and mainly gastrointestinal symptoms. Overall, no life-threatening reaction was observed throughout the administration of Berberine compounds [70],[68].”

13.The title of table 2 and 4 should be described completely so that people can understand the content in table from the title.

Response: We accordingly renamed these tables to be more understandable.

14.How were the pathways or molecules changed in table 2-4 ï¼Ÿ Up-regulated or down-regulated, activated or inhibited ? Please mark.

Response: We added this information to the table, and this is highlighted in the revised manuscript. Also, we would like to mention that we have a very comprehensive table that will be submitted as a supplementary table, due to the fact that it is very large.

Pathway

Liver and spleen microbiota cultures.

Genomic DNA analysis for gut microbiota.

Genomic DNA analysis for gut microbiota

ELISA for inflammatory cytokines levels.

Increased expression and analysis of multiple cellular pathways involved in NAFLD induction and progression.

Up-regulation of FXR pathway and FGF-15 levels.

Genomic DNA analysis for gut microbiota.

Upregulation of Adiponectin receptor 2 (AdipoR2) levels and mitogen-activated protein kinase (MAPK) ERK expression and their influence on inflammation and progression of NAFLD.

Activation of AMPK pathway

Genomic DNA analysis for gut microbiota.

ELISA for inflammatory cytokine levels

Downregulation of nuclear translocation of NF-κB via the TLR4/MyD88/NF-κB pathway.

Downregulation of chemerin/CMKLR1 signalling pathway and Treg/TH17 ratio.

Up-regulation of both mitochondrial activity and AMPK pathway,

Genomic DNA analysis for gut microbiota

Down-regulation of triglyceride transfer protein (MTTP), apolipoprotein B, and low-density lipoprotein receptor (LDLR)

Upregulation of adipocyte macrophage-derived Angptl2 signaling pathway analysis.

Upregulation of NOD1, NOD2 pathway

NLRP3 inflammasome activity

Activation of SIRT3/AMPK/ACC pathway

Activation of SIRT3 pathway and its effects on lipid metabolism.

Stimulation of mitochondrial β-oxidation levels.

Micro 18F-FDG PET/CT

Temperature and metabolic activity measurements.

Fat adipose tissue analysis.

Downregulation of p38MAPK/ERK-COX2 pathway. analysis

Upregulation of Nrf2/ARE signaling pathway and genes involved in the inflammatory response.

Activation of AMPK pathway

 Downregulation of sterol regulatory element-binding protein 1c (SREBP-1)c, pERK, TNF-α, and pJNK- genes involved in inflammatory response

Liver enzyme profile and serum lipid profile.

Histological analysis.

Activation of C‑X‑C chemokine receptor type 4 (CXCR4)/CXCL12 signaling pathway.

ELISA for detecting proinflammatory cytokine levels. (TNF‑α and IL‑6).

Upregulation of PPAR‑γ and Insulin receptor (IR) activation

Liver enzyme profile and lipid profile

Histological analysis.

RT-PCR for key enzymes and protein

Downregulation of lipogenic genes: fatty acid synthase (FAS), stearoyl-CoA desaturase (SCD1), and sterol regulatory element-binding protein 1c (SREBP1c)

Upregulation of lipolytic gene carnitine palmitoyltransferase-1 (CPT1)

Upregulation of peroxisome proliferator-activated receptor γ (PPARγ) in adipose tissue and the liver

Downregulation of phosphatidate phosphohydrolase (PAP) activity

Antioxidants level  assay

Stimulation of mitochondrial activity and reactive oxygen species levels.

Upregulation of SIRT3 activity level measurements.

Downregulation of uncoupling protein-2 (UCP2) mechanism.

Significant upregulation of insulin receptor substrate-2 (ISR-2) levels analysis.

Downregulation of genes associated with lipid metabolism: CPT-1, MTTP, and LDLR.

15.There is no a clear or specific conclusion for mechanism of berberine effect on NAFLD in the results, or what’s the main mechanism behind ? what’s the regulation network?

Response: As you suggested, we added a figure that illustrates the main pathways in which Berberine intervention acts NAFLD.

Figure 4

Figure 4. The main Berberine modulation pathways in NAFLD

Legend

NRLP3- NACHT, LRR, and PYD domains-containing protein 3 inflammasome, ROS- reactive oxygen species, Nrf/HO-1- Nrf2- nuclear factor erythroid -related factor 2 /Heme Oxygenase 1– related factor, FAO- fatty acids oxidation, Nrf/ARE- nuclear factor erythroid -related factor/antioxidant response element, SIRT1- Sirtuin 1, SIRT3- Sirtuin 3, TLR4/NF-kB-p65-  toll-like receptor 4/ nuclear factor kB-p65, AMPK- Adenosine Monophosphate-Activated Protein Kinase, SREBP-1c- sterol regulatory element-binding protein 1c, BBR- berberine, PPAR-γ -peroxisome proliferator-activated receptor-γ ( activation mechanism, Ë« downregulation, --- effect) (Created with Biorender.com)

16.The results part looks like a paper list of berberine in vivo and in vitro work. The content should be enriched.

Response: We added content to this section, but the main information is found in the discussion section.

“Recent studies conducted on both animal model experiments as well as different types of cell cultures showed that BBR not only managed to modulate lipid metabolism but also has anti-steatotic and anti-inflammatory effects. The studies on various experimental models described BBR effects on multiple pathways involved in the progression to NAFLD, and the major ones comprise activation of the SIRT3 pathway, upregulation of SIRT1, activation of AMPK pathway, suppression of NLRP3 pathway, and act as PPAR-ϒ agonist. (Figure 4)”

  1. In section “Pathophysiology and Molecular Signaling”, what’s the regulation network for those molecules mentioned in this section?

Response: We highlighted the upregulation or inhibition mechanisms in this section, and the regulation network may also be found in Figure 4.

“Furthermore, BBR, by activating the AMPK pathway, proved to downregulate the expression of SREBP-1c both in vivo and in vitro [127],[141],[97],[102]. SREBP-1c (transcription factor sterol regulatory element binding protein-1c) is a key protein that promotes a series of genes responsible for fatty acid production and genes involved in glucose homeostasis [150]”

“Another sirtuin on which recent studies showed that BBR activates is Sirtuin 1 (SIRT1). Wang P. et al. and Rafiei H. et al. proved that by upregulating SIRT1 expression, BBR managed to ameliorate changes induced by NAFLD, stimulated fatty acid oxidation, decreased intracellular lipid accumulation, as well as modulated mitochondrial respiratory subunits”.

“Recent studies show that BBR action on mitochondria proves to be mediated by various pathways. Xu X. et al. found that via SIRT3, mitochondrial β-oxidation was enhanced in BBR-treated mice. Therefore, a better control of lipid metabolism was obtained [98].  Moreover, Sun Y. et proved that BBR via downregulation of Nrf2/HO-1 pathway expression managed to decrease reactive oxygen species levels by particularly influencing I and III respiratory chains[130].

Regarding its anti-inflammatory properties, Teodoro JS et al. further demonstrated that by upregulating SIRT3 expression, BBR completely restored mitochondrial activity[110].”

“By inhibiting the mitogen-activated protein kinase (MAPK) pathway, BBR blocked the inflammation and angiogenesis associated with NAFLD in HFD-induced mice and showed potential anti-tumor effects”.

18.Some content about berberine found from the analysis is suggested to be included in the results rather than discussion part.

Response: As you well suggested we added some content about Berberine to the Results section.

“The Berberine action in humans is related to its biological availability, which greatly influences the results. According to the Biopharmaceutics Classification System (BCS), berberine could be classified as BCS Class II and IV. This classification is attributed to berberine's biological effects being influenced by its bioavailability, which remains lower than 0.01 within 48 hours. This limitation is primarily due to its restricted solubility and slow dissolution in water[77].

Also, berberine has decreased absorption in the intestines, showing that less than one-third is absorbed at the intestinal level in animal studies [78].

This is an impediment because the easiest and optimal administration is the oral route, the parenteral administration being related to adverse toxicity of berberine, uneven diffusion, low cellular internalization, and fast drug elimination [78], [79]. Another factor that affects the berberine’s biological availability is the intervention of efflux membrane transporter P-glycoprotein, which limits the cellular uptake of the xenobiotics in the gastrointestinal tract[77].

The methods developed to increase berberine’s availability are delivery by nanosized carriers with both inorganic and organic components such as polymers, lipids, gold, or magnetic porous nanomaterials [78]. In the study of Kohli K. et al., the non-toxic, biologic-compatible polymers used were chitosan and alginate, and the results showed a remarkably increased bioavailability of berberine [77]. The nanotechnology medication formulation with encapsulating micelles as transporters enables the solubilization of hydrophobic BBR molecules without altering the pharmacological composition [80].  BBR as a hydrophobic core enveloped in a thin shell of hydrophilic micelle segments  (BBR-incorporated micelle solution) proved to be feasible with an 80% rate of success and an increased effect of BBR compared with standard formulation administration in preclinical studies [81]. Another proposed formula used poly-lactic acid (PLA) as a bioactive material configuration stabilized by the coaxial electro-spraying procedure for obtaining  BBR nano-layering [82].  In this in vivo and in vitro study, the results showed a better internalization of BBR-loaded poly-lactic acid nanoparticles at the cellular level compared with standard BBR, with a better release and action on neoplastic transformed cells [82].

Other proposed delivery systems to improve the bioavailability and to decrease the degradation of BBR were carriers such as lipid vesicles, non-ionic surfactant vesicles, dendritic nanoparticles, nanosilver or nanogold particles, ultrafine emulsions or ethosomes ( ethanol-based particles) [83]. “

Sincerely yours,

Sandica Bucurica

Reviewer 2 Report

Comments and Suggestions for Authors

The review deals with the efficacy of berberine in treating NAFLD. The literature search on this topic is comprehensive, covering recent relevant studies. While the Figures and the Tables are well-designed, the review is very hard to read and in some parts confusing. To my opinion, there are far too many, often disjointed paragraphs, which distracts from the content and finally limits the reading flow.

For Chapter 1.1., I would suggest starting with the pathology before the exceptions in diagnosis are mentioned. Additionally, the scoring system should be presented more comprehensively without providing a list of individual data. Before Chapter 2, the basics and hypotheses of the work have to be explained. What is the rationale for the approach?  Chapter 4 is not a discussion of the results which are on berberine related to NAFLD.  However, the part on the pathophysiology and molecular signalling in NAFLD is very well written, but disconnected from the context.

The article suffers from poor writing quality and numerous inaccuracies, which detract from its content. A thorough revision is necessary to enhance its readability. Reorganizing the chapters and proofreading are recommended to enhance the quality of the article.

Minor comments:

Are there more recent data on the increase of MAFLD?  Could one extrapolate the increase rate from the prevalence data, which are from reports 2022/2023? The dates of the studies should be mentioned in the text.

Why is PET/CT not described as an imaging technique?

Line 67:  Please, use the abbreviation NAFLD.
Line 102: Please, explain what you mean by (Almaza).
Line 180: Please, explain the term AUROC and its meaning.
Line 182: Add score.
Line 186: Please, start with Figure 1, which could be referenced at the beginning of the chapter.
Lines 224/226: NIS-4 is not a biomarker but a blood-based diagnostic test.
Lines 229/233: The diagnostic tests need to be explained more clearly. Please, rephrase.
Line 258:  BBR is berberine.
Line 278/280: Please, check.
Line 526: Bracket is missing.
Line 548/550: Reference is missing.
Line 572: Two typos.
Line 574/575: Check the meaning of the sentence. And correct efficacity.
Line 652: cand
Line 657: form
Line 662: obtained
Line 669: in

I think there are many more errors in the article that I have overlooked.

Comments on the Quality of English Language

The article needs proofreading to increase its readability. 

Author Response

Dear Reviewer,

Thank you for giving us the opportunity to improve our  manuscript “Berberine and  pre-fibrotic stages of non-alcoholic fatty liver disease - clinical and preclinical  overview and systematic review of the literature

            We appreciate the time and effort you dedicated to providing feedback on our manuscript, and we are grateful for your insightful comments and valuable improvements to our paper.

We have incorporated your highly valuable suggestions, highlighted them in the manuscript, and also reorganized the chapters to be more comprehensible, clear, and readable.

The review deals with the efficacy of berberine in treating NAFLD. The literature search on this topic is comprehensive, covering recent relevant studies. While the Figures and the Tables are well-designed, the review is very hard to read and, in some parts, confusing. To my opinion, there are far too many, often disjointed paragraphs, which distracts from the content and finally limits the reading flow.

  1. For Chapter 1.1., I would suggest starting with the pathology before the exceptions in diagnosis are mentioned. Additionally, the scoring system should be presented more comprehensively without providing a list of individual data.

Response: We removed the individual data and added the score

“The Youden used cut-offs varied between 302dB/m for grade 1 and 337 dB/m for grade 3 of steatosis [30]. The CAP score grades steatosis as grade 0 for less than 278 dB/m, between 278-301dB/m for grade 1, 303-337 for grade 2, more than 337 for grade 3. (Figure 2)”

  1. Before Chapter 2, the basics and hypotheses of the work have to be explained. What is the rationale for the approach? 

Response:

“The aim of our review is to highlight berberine’s effects as a natural and available compound on different metabolic processes, with emphasis on its different molecular pathways and their outcomes regarding NAFLD.”

The actual available therapies for NAFLD are still under development and need to be tested. As a result, there is a pressing need to develop new therapeutic strategies to address this unmet need.

On the other hand, berberine is a natural compound used for many years as an accessible, versatile herbal therapy, especially in oriental medicine. It has proven hypolipemic and anti-steatotic effects in NAFLD [9], [10].”

Lifestyle changes are challenging at the individual level and require long-term upholding, necessitating the development of medical treatments [58].

The main reason for the delayed development of medical treatment for NAFLD is the difficulty in pairing an accurate preclinical experimental prototype to resemble the human liver, most used for experimental studies being mice, rats, cell cultures, and guinea pigs, which are the most feasible [9].”

  1. Chapter 4 is not a discussion of the results which are on berberine related to NAFLD.  However, the part on the pathophysiology and molecular signaling in NAFLD is very well written, but disconnected from the context.

Response:

We added the Pathophysiology to introductions section, as we reorganized the chapters.

  1. The article suffers from poor writing quality and numerous inaccuracies, which detract from its content. A thorough revision is necessary to enhance its readability. Reorganizing the chapters and proofreading are recommended to enhance the quality of the article.

Response: We reorganized the chapters to be a more clear and readable manuscript, and we thank you for your attention and well pointed suggestions.

  1. Are there more recent data on the increase of MAFLD?  Could one extrapolate the increase rate from the prevalence data, which are from reports 2022/2023? The dates of the studies should be mentioned in the text.

Response:

We added this data  to the manuscript “Due to the increasing prevalence of obesity and diabetes, it is projected that the burden of MAFLD will rise in the next decade. A recently published study in 2024 emphasizes the fact that epidemiological data changes when considering the prevalence of MAFLD (which is a more inclusive diagnosis) versus NAFLD and finds at least 5%-10%  more MAFLD worldwide reported (30-40%) compared with NAFLD reports of 25-30% [8].” 

  1. Why is PET/CT not described as an imaging technique?

Response: we added the information about PET/CT

“In regards to Positron Emission Tomography - Computed tomography scan  (PET/CT) usefulness in staging and diagnosing NAFLD, there are needed standards because the relationship between body mass index and correction of standardized uptake value (SUV) of [18F]2-fluoro-2-deoxy-D-glucose (18F-FDG)  dosage is not established in non-malignant cases [35], [36]. Alternative PET/CT tracers proposed for liver steatosis diagnosing are  14(R,S)-[18F]fluoro-6-thia-Heptadecanoic acid (18F-FTHA), 11C-Palmitate and 11C-Acetate, but still need validation and standardization [37]. “

  1. Line 67:  Please, use the abbreviation NAFLD.-Corrected
    Line 102: Please, explain what you mean by (Almaza).- corrected
    Line 180: Please, explain the term AUROC and its meaning.corrected
    Line 182: Add score.

Score added The Youden used cut-offs varied between 302dB/m for grade 1 and 337 dB/m for grade 3 of steatosis [30]. The CAP score grades steatosis as grade 0 for less than 278 dB/m, between 278-301dB/m for grade 1, 303-337 for grade 2, more than 337 for grade 3.(Figure 2)
Line 186: Please, start with Figure 1, which could be referenced at the beginning of the chapter.

We started with the figure, which has been added to the beginning of the section
Lines 224/226: NIS-4 is not a biomarker but a blood-based diagnostic test.

Corrected and rewrote the whole chapter of serum markers
Lines 229/233: The diagnostic tests need to be explained more clearly. Please, rephrase.

The content was modified to be more clear.
Line 258:  BBR is berberine.- corrected
Line 278/280: Please, check.- corrected
Line 526: Bracket is missing. Corrected
Line 548/550: Reference is missing. The reference was inserted
Line 572: Two typos. Corrected
Line 574/575: Check the meaning of the sentence. And correct efficacity.
Line 652: cand- corrected
Line 657: form- corrected
Line 662: obtained- corrected
Line 669: in- corrected
We look forward to hearing from you in due time regarding our submission and to respond to any further questions and comments you may have.

Sincerely yours,

Sandica Bucurica

Round 2

Reviewer 1 Report

Comments and Suggestions for Authors

The major concern is the limited content of berberine in results part and novelty,  there are several good and comprehensive reviews published about this topic (doi: 10.3390/nu14173459,10.1155/2013/308134,10.1155/2016/3593951 etc.).

Comments on the Quality of English Language

 Minor editing of English language required.

Author Response

Dear Reviewer,

 We appreciate your comments, and your suggestions contributed to improving our manuscript.

The strength and novelty of our manuscript lie in its comprehensive approach. It is a systematic review that covers both clinical and preclinical studies, providing a thorough examination of Berberine’s effects on NAFLD. Most of the literature we found highlights the beneficial effects of BBR on metabolic and molecular reactions involved in both lipid and glucose metabolism.

 What’s more, recent studies that we researched in our review proved that this natural compound not only modulates lipid metabolism but also targets mitochondrial activity, inflammatory cascade, and bacterial translocation. By doing so, BBR acts on the main pathophysiological mechanisms involved in NAFLD.

 We would like to bring forward new data that we collected from the studies that we analyzed. In our systematic review, we included novel results regarding Berberine’s usage in NAFDL and its beneficial effects on this category of patients; thus, we encompass recent studies regarding BBR treatment in steatotic liver disease. Even though most of the evidence-based on both human and animal experiments needs further testing, we hold a strong belief that BBR opens a new perspective for NAFLD’s treatment.

Reviewer 2 Report

Comments and Suggestions for Authors

The authors have significantly improved the manuscript. It can be accepted as it is. 

Author Response

Dear Reviewer,

We are grateful for your recommendation and your suggestions helped us to have an improved manuscript.

Sincerely yours,

Sandica Bucurica

Round 3

Reviewer 1 Report

Comments and Suggestions for Authors

Please take time to check and improve the manuscript carefully.

1. The sections are not continuous. Sections 1.1.1 , 1.2 are not found.

2. The format including line spacing especially in figure legend should be consistent.

3. There are some short one sentence paragraphs are suggested to be combined with other paragraphs especially when same topic is discussed.

4. Please optimize the keyword list, non-alcoholic fatty liver disease and NAFLD should be a keyword non-alcoholic fatty liver disease(NAFLD), and MAFLD is also important.

5. "MAFLD" was introduced to replace "NAFLD", but "NAFLD"  was still used in the whole manuscript.

6. As commented previously, the results are a list of publications in vivo and in vitro. Publications are also suggested to be sub-grouped according to the mechanisms listed in Discussion.

7. Please include the strength and novelty of your manuscript in the conclusion.

8. The introduction seems too long and should be refined.

Comments on the Quality of English Language

Minor editing of English language required.

Author Response

Dear Reviewer,

We appreciate very much the consistent effort that you dedicated to help us improve our paper, and we incorporated your suggestions and reorganized the manuscript. Thank you for giving us this opportunity

  1. The sections are not continuous. Sections 1.1.1 , 1.2 are not found.

We corrected accordingly.

  1. The format including line spacing especially in figure legend should be consistent.

The line spacing will be adjusted during journal  editing, but for the time being this is the possibility of editing  permitted by the template.

  1. There are some short one sentence paragraphs are suggested to be combined with other paragraphs especially when same topic is discussed.

We combined the sentences with the corresponding paragraphs.

  1. Please optimize the keyword list, non-alcoholic fatty liver disease and NAFLD should be a keyword non-alcoholic fatty liver disease(NAFLD), and MAFLD is also important.

Indeed, we missed mentioning MAFLD as a keyword, and we added it as you well suggested.

  1. "MAFLD" was introduced to replace "NAFLD," but "NAFLD"  was still used throughout the manuscript.

The studies that we reviewed referred to NAFLD, and since the term MAFLD was proposed just recently, we respected the studies' terminology of NAFLD.

  1. As commented previously, the results are a list of publications in vivo and in vitro. Publications are also suggested to be sub-grouped according to the mechanisms listed in the Discussion.

We inserted content into the results section, but the rationale of our tables respects the publication year of every research article we included in our review. The publications are discussed and grouped according to the mechanism in every sub-section of the Discussion section. Also, we enriched the results section “As we found BBR plays a complex role when it comes to modulating different metabolic processes and through the studies that we analysed it was shown that that BBR has also different pharmacological effects. The existent literature consists of different experimental models ranging from different species of rats and mice to various cell cultures. When we talk about animal experiments, BBR proved to reduce anthropomorphic features of rodents, improved control on lipid and glucose metabolism as well as targeting inflammation by modulatingn different genes and pathways involved in stress oxidative response. Moreover, in cell cultures BBR proved to have an effective anti-steatotic property shown as a decrease in accumulation of lipids or as an improvement of NAFLD-induced changes in different cell lines. In the last years researchers were much keener on finding the relationship between different inflammatory pathways and their contribution to NAFLD’s progression. [152] Some of the studies that we analysed found that mitochondria may play a bigger role in stress oxidative response than it was known. Xu X. et al. (2019), Sun Y. et al. (2017), Rafiei H. et al (2017) and Teodoro JS. et al. (2013),  all proved through their research that BBR modulated important mitochondrial respiratory complex subunits especially I and III mitochondrial respiratory chains and by doing so managed to decrease the levels of ROS and lower the inflammation both in vivo and in vitro. [98], [110], [130], [144]

In order to have a better understanding of NAFLD’s pathophysiology is necessary to assess how changes of different species of gut microbiota promotes fat liver accumulation. [45], [151], [127].  Genomic DNA analysis of gut microbiota proved to be a field of interest for Chen D. et al (2023), Zhou LM. et al. (2023), Dai Y. et al (2022), Yang S. et al (2022), Li H. et al (2022), Cossiga V. et al (2021) and Shu X. et al (2021), , who managed to reveal that BBR not only blocks bacterial translocation but also rebalances gut microbiota by stimulating beneficial bacteria in animal model experiments such as Atopobiaceae, Brevibacterium, Christensenellaceae, Coriobacteriales, Papillibacter, PygmaiobacterRikenellaceae RC9 and Prevotella in animal model experiments. [84], [85], [87],[90], [118], [121], [122]”

Another relative new pathway that seems to be involved in NAFLD’s pathophysiology that has been studied recently is the one related to Clock and Bmal1 genes that are involved in circadian rhythm control. [156] Ye C. et al. (2023) found an interesting association between these genes and BBR treatment. These genes are important for metabolic and redox homeostasis, thus the disruption of their activity acts as a trigger for inflammatory response. After BBR treatment by downregulating the activity of Bmal1 and Clock genes the inflammation was lowered and thus the metabolic balance of different molecular reactions was restored. [111], [115]. With aid form all of these new findings in the field of NAFLD pathology, there is hope for finding a suitable treatment that could prevent the many complications associated with this disease.

  1. Please include the strength and novelty of your manuscript in the conclusion.

We added the strength and novelty of our manuscript in the conclusion. The strength and novelty of our review lie in its comprehensive approach. It is a systematic review that covers both clinical and preclinical studies, providing a thorough examination of Berberine’s effects on NAFLD. Most of the literature we found highlights the beneficial effects of BBR on metabolic and molecular reactions involved in both lipid and glucose metabolism. What is more, recent studies that we researched in our review proved that this natural compound not only modulates lipid metabolism but also targets mitochondrial activity, inflammatory cascade, and bacterial translocation. By doing so, BBR acts on the main pathophysiological mechanisms involved in NAFLD.

  1. The introduction seems too long and should be refined.

We adjusted the Introduction section, but we still consider that relevant information regarding the NAFLD-wide subject cannot be left out.

Sincerely yours,

Sandica Bucurica

Round 4

Reviewer 1 Report

Comments and Suggestions for Authors

Although the editorial board will help to format the manuscript during journal  editing, it is suggested that the authors also make effort to do it before submit so that manuscript can be double checked.

Comments on the Quality of English Language

Minor editing of English language required.

Author Response

Dear Reviewer,

Your suggestions were very helpful, and we are grateful for them. We rechecked the format and edited where appropriate, paying attention also to English editing.

The collective of authors.